# Benthic foraminiferal Mn/Ca ratios reflect microhabitat preferences

Karoliina A. Koho[1,2], Lennart J. de Nooijer[3], Christophe Fontanier[4,5,6], Takashi Toyofuku[7], Kazumasa Oguri[7], Hiroshi Kitazato[7,8], and Gert-Jan Reichart[2,3]

[1]Department of Environmental Sciences, P.O. Box 65 (Viikinkaari 1), 00014 University of Helsinki, Finland
[2]Department of Earth Sciences – Geochemistry, Faculty of Geosciences, Utrecht University, P.O. Box 80.021, 3508 TA Utrecht, The Netherlands
[3]Department of Ocean Systems, NIOZ-Royal Netherlands Institute for Sea Research and Utrecht University, Den Burg, The Netherlands
[4]Ifremer, Géosciences marines, Z.I. Pointe du Diable CS10070, F29280 Plouzané, France
[5]Univ. of Bordeaux, Environnements et Paléo-environnements Océaniques et Continentaux, UMR 5805, F33600 Talence, France
[6]Univ. of Angers, F49035 Angers, France
[7]Japan Agency for Marine-Earth Science and Technology (JAMSTEC), 2-15 Natsushima-cho, Yokosuka, 237-0061, Japan
[8]Tokyo University of Marine Science and Technology, 4-5-7 Konan, Minato-ku 108-8477, Tokyo, Japan

*Correspondence to*: Karoliina A. Koho (karoliina.koho@helsinki.fi)

**Abstract.** The Mn/Ca of calcium carbonate tests of living (rose Bengal stained) benthic foraminifera (*Elphidium batialis*, *Uvigerina* spp., *Bolivina spissa*, *Nonionellina labradorica* and *Chilostomellina fimbriata*) were determined in relation to pore water manganese (Mn) concentrations for the first time along a bottom water oxygen gradient across the continental slope along the NE Japan margin (western Pacific). The local bottom water oxygen (BWO) gradient differs from previous

field study sites focusing on foraminiferal Mn/Ca and redox chemistry, therefore allowing further resolving previously observed trends. The Mn/Ca ratios were analyzed using laser ablation ICP-MS, allowing single-chamber determination of Mn/Ca. Incorporation of Mn into the carbonate tests reflects environmental conditions and is not influenced by ontogeny. The inter-species variability in Mn/Ca reflected foraminiferal in-sediment habitat preferences and associated pore water chemistry, but also showed large interspecific differences in Mn partitioning. At each station, Mn/Ca ratios were always

lower in the shallow infaunal *E. batialis,* occupying relatively oxygenated sediments, compared to intermediate infaunal species, *Uvigerina* spp. and *B. spissa,* which were typically found deeper, under more reducing conditions. The highest Mn/Ca was always recorded by the deep infaunal species *N. labradorica* and *C. fimbriata*. Our results suggest that although partitioning differs, Mn/Ca ratios in the intermediate infaunal taxa are promising tools for paleoceanographic reconstructions as their microhabitat exposes them to higher variability in pore water Mn, thereby making them relatively sensitive recorders

of redox conditions and/or bottom water oxygenation.

## 1 Introduction

Benthic foraminifera, single-celled, testate eukaryotes, are common proxies used in paleoceanographic studies. Many species make a shell, or a test, of calcium carbonate that has a high preservation potential. The chemistry of the carbonate test (i.e. its

isotopic and elemental composition) reflects various physical and chemical conditions of the calcification environment, thereby allowing reconstruction of past environmental and climatic conditions. One of the most commonly applied geochemical foraminifera-based proxies is the Mg/Ca ratio of the test carbonate, which has been shown to primarily reflect seawater temperatures (e.g. Nürnberg et al. 1996, Elderfield et al. 2006). Other elemental ratios, such as B/Ca and U/Ca, have been shown to reflect carbonate chemistry (e.g. Yu and Elderfield 2007, Yu et al. 2010, Keul et al. 2013). Previous studies have also highlighted the potential of reconstructing bottom water oxygenation (BWO) and/ or sediment redox chemistry, using Mn/Ca ratios in benthic foraminifera (Reichart et al. 2003, Glock et al. 2012, Groeneveld and Filipsson 2013, Koho et al. 2015, McKay et al. 2015). The relationship between Mn incorporation into foraminiferal test carbonate and oxygenation is based on the combination of Mn availability and redox chemistry, which is typically linked to BWO. Under oxic conditions Mn is present in form of solid (hydr)oxides, i.e $MnO_2$ or $MnOOH$, on coatings on sediment particles (e.g. Finney et al. 1988). Therefore, foraminifera calcifying under oxic condition are likely to incorporate no, or very low amounts of Mn (Koho et al. 2015). In contrast, in the absence of oxygen the solid Mn-(hydr)oxides are reduced to aqueous $Mn^{2+}$ (Froelich et al. 1979), subsequently leading to build up of bio-available $Mn^{2+}$ in pore water. Foraminifera calcifying under such conditions are expected to show elevated Mn/Ca ratios, the concentration depending on the actual in situ aqueous Mn-concentrations (Munsel et al. 2010). Exceptions are environments, like oxygen minimum zones, where bottom waters have been oxygen-deprived for extended periods. In such cases, aqueous $Mn^{2+}$ has diffused upwards and was released into the overlying water, leaving pore waters (and sediments) depleted in $Mn^{2+}$ (e.g. Van der Weijden 1998, Law et al. 2009). In such settings, foraminiferal test calcite is expected to contain no Mn (Koho et al. 2015). In sediments, where bottom waters and surficial sediments are oxygenated and deeper sediments are anoxic, and hence Mn is retained in sediments, incorporation of Mn in different species of foraminifera is expected to depend on the species-specific in-sediment living depth.

Benthic foraminifera are traditionally divided into four categories based on their microhabitat: epifauna, shallow-, intermediate- and deep infauna (Corliss 1985, Jorissen et al. 1995). This depth distribution is tightly controlled by species-specific responses to environmental redox chemistry and food supply (e.g. Jorissen et al. 1995, Koho et al. 2008, Koho and Piña-Ochoa 2012). Epi- and infauna, living above the sediment water interface and in surficial sediments respectively, are typically found under oxic conditions with increasing living depth corresponding to increasing oxygen depletion and redox stress (e.g. Koho et al. 2008, Koho and Piña-Ochoa 2012). Therefore, test chemistry of species with different microhabitat preferences, i.e. living and/or calcifying at different sediment depths with varying Mn redox chemistry, are expected to display different Mn/Ca. This was confirmed in a study of Koho et al (2015), showing that shallow infaunal species consistently had lower Mn/Ca ratios than species living at the same location, but found deeper in the sediment. This resulted in a conceptual model linking bottom water oxygenation, organic matter supply and microhabitat effects (Koho et al., 2015). However, this model is currently based on a limited set of oceanic conditions and theoretical considerations.

Here we present Mn/Ca ratios in benthic foraminifera with various microhabitat preferences collected from a depth transect across a dysoxic-to-oxic zone (BWO always ≥33μmol/L) in northern Japan. The incorporation of Mn into foraminiferal test

carbonate is evaluated in the context of microhabitat distributions and foraminiferal ecology, which is described previously (Fontanier et al. 2014) for this area, and compared against measured bottom- and pore water chemistry.

## 2 Materials and Methods

### 2.1 Study Area

The sampled transect is located at the continental slope of NE Japan, off Hachinohe (Fig. 1). Surface waters in the area are dominated by three major currents: the Tsugaru Warm Current, Kuroshio Current and Oyashio Current. The convergence of these current systems results in a number of hydrological fronts sustaining high productivity in this area (Saino et al. 1998, Itou et al. 2000). Below 200m water depth, the North Pacific intermediate Waters (NIPW) mixes gradually with saline Deep Pacific Water (DPW), entering this area between a water depth of 800-3000 m. The development of a dysoxic water mass
between water depths of 700-1500 meters is related to both high surface water productivity, resulting in enhanced remineralization of organic matter and associated oxygen consumption, and poor intermediate water ventilation at depth (Nagata et al. 1992).

### 2.2 Sampling

Sediment samples were collected in August 2011 onboard of R/V *Tansei Maru* (Atmosphere and Ocean Research Institute,
University of Tokyo/JAMSTEC) with a Barnett-type multi-corer equipped with eight Plexiglas tubes with an internal diameter of 82 mm (Barnett et al. 1984). This type of coring device allows recovery of undisturbed sediments with an intact sediment-water interface. Sediment for faunal analyses was collected over a transect spanning the OMZ, whereas pore water chemistry was determined from material collected at 3 selected sites within this transect (Fig. 1, Table 1). Separate cores were collected for pore water- and foraminiferal analyses, and oxygen profiling, all of which were derived from the same
multicore cast. In addition to coring, a Conductivity-Temperature-Depth (CTD) cast (Sea-Bird Electronics, S/N 860; SBE9plus) equipped with SBE3 thermometer (S/N 4378), SBE4 conductivity sensor (S/N 3307) and SBE43 oxygen sensor (S/N 0781) was taken at every site to record water column properties. The accuracy specifications of the oxygen sensor are typically within 2% of true value.

### 2.3 Pore water analyses

Immediately upon arrival on board, bottom water samples were taken from overlying multi-core water after which the core was transferred via a table with a closely fitted hole into a $N_2$-purged glove bag for sequential slicing (atmospheric $O_2$ within the glove bag never exceeding 1%). The core was subsequently sliced down to 20 cm depth: the first two centimeters with a resolution of 0.5 cm intervals, between two to ten centimeters, samples were taken at 1 cm intervals and from ten centimeter downward with 2 cm intervals. Sediment samples were centrifuged in 50 ml tubes for 20 minutes at 2800 rpm. The
supernatant was removed and filtered over 0.45 μm TeflonTM filters under $N_2$ atmosphere, and divided into subsamples for

various analyses. The nutrient samples were stored at -20°C until analyses, and back in the laboratory nitrate concentrations were measured with a Bran-Luebbe AA3 autoanalyser and ammonium spectrophotometrically using phenol-hypochlorite (Helder and De Vries, 1979). Samples for pore water elemental analyses were acidified with suprapur HCl 37% (10μl per ml of sample) and subsequently stored at 4°C until analyses at Utrecht University. Seawater elemental concentrations of [55]Mn were measured with an inductively coupled plasma-mass spectrometer (ICP-MS, ThermoFisher Scientific Element2-XR). Replicate analyses and an in-house standard indicated that the relative error for analyses of pore water element concentrations was generally <3%.

## 2.4 Oxygen microprofiles

The oxygen microprofiles were recorded in a custom-built incubation chamber, allowing regulation of temperature and oxygen content of overlaying water, and have been previously published in Fontanier et al (2014). Here in short: upon retrieval on board, one core (from stations 6,8 and 10 only) was immediately subsampled with a piston device made of a 50 ml syringe, and subsequently placed into the incubation chamber filled with bottom water collected with Niskin bottles. Every core was left to stabilize for minimum of nine hours, while the temperature and oxygen concentrations were kept at *in situ* conditions. Any fluctuations in the oxygen concentrations were less than 0.5 μmol/l. After the oxygen microprofiles were measured with an OX-50 Unisense microsensor and a motor controller (step size of 100 μm).

## 2.5 Foraminifera: sampling and elemental composition

The details of benthic foraminiferal processing and analyses are described in Fontanier et al. (2014). In summary, cores for faunal analyses were sliced at 0.5 cm intervals down to 4 cm, from four to six centimeter depth at 1 cm interval and at 2 cm intervals down to 10 cm depth in sediment. Samples were preserved and stained with rose Bengal dissolved in 95% ethanol (1g/L). Stained (living) foraminifera in the >150 μm fraction were wet picked, identified and stored on micropaleontological slides. From the census data of Fontanier et al (2014), the average living depth (ALD) of selected species was calculated based on the equation in Jorissen et al (1995).

Few species occurring in high relative abundance and representing various microhabitats were selected for Mn/Ca measurements (Table 2). Most of the specimens came from surficial sediments (0-0,5 cm) but for some taxa, specimens from deeper sediment intervals were measured additionally (Table 2.) Prior to analyses, all foraminifera were thoroughly cleaned to remove sediment contamination (Barker et al. 2003) by placing the foraminifera in Eppendorf tubes and rinsing them 3 times in ultrapure water (100μl). This was followed by 3 rinses in methanol (100μl), and finally 3 more rinses in ultrapure water (100μl). Between the methanol rinses foraminifera were placed in an ultra sonic bath for approximately 5 seconds. After these steps, specimens were dried and stored until geochemical analyses.

Trace element content was measured generally on single foraminiferal chambers (Table 2) with two different laser ablation ICP-MS set ups, which have been shown to produce comparable foraminiferal elemental/Ca results (de Nooijer et al. 2014). In all cases, shells were ablated from the outside towards the inside (Fig. 2) in He environment and element ratios were

based on averaging measured concentrations during each ablation after selecting the non-contaminated part of the ablation profile, which was recognized by elevated counts of Al, Mg, Mn at the beginning, and occasionally end of, the ablation profile (Fig. 2). Although all tests were carefully cleaned, test surfaces of foraminifera can still be contaminated with adhered particles containing elevated concentrations of Mg and Mn in combination with elevated Al. These parts of the ablation profiles were excluded from further consideration (Fig. 2). Short ablation profiles (generally <5s) were excluded from the data, and only longer ablation profiles, typically ranging between 10-30 seconds in length, were used.

Measurements carried out at Utrecht University were done with a deep-ultraviolet wavelength laser (193 nm) using a Lambda Physik excimer system with GeoLas 200Q optics (Reichart et al., 2003). Every ablation lasted approximately 130 seconds, of which the first 45 seconds consisted of a background. Ablation craters were circular with a diameter of 80 μm, pulse repetition rate was 5 Hz and the energy density at the sample surface approximately 1 J/cm$^2$. Element to calcium ratios were quantified using counts for $^{27}$Al, $^{43}$Ca, $^{44}$Ca, $^{24}$Mg, $^{26}$Mg and $^{55}$Mn and their relative natural abundances on a sector field-ICP-MS (Element2, Thermo Scientific). The cycle length through all masses was 0,64 seconds. Raw counts were converted to element concentrations and integration windows were set using the computer program Glitter (developed by the ARC National Key Centre for Geochemical Evolution and Metallogeny of Continents (GEMOC) and CSIRO Exploration and Mining). Calibration was performed against international NIST SRM 610 glass standard (using concentrations from Jochum et al., 2011) at a higher energy density (5 J/cm$^2$), which was ablated twice every 12 samples. Calibration of element/calcium ratios in calcium carbonate samples using a NIST glass standard has been demonstrated to be accurate for many elements when using a 193 nm laser (Hathorne et al., 2008). Switching energy density between carbonate sample and glass standard has been shown not to affect the concentration of the relevant elements (Dueñas-Bohórquez et al., 2011).

Some samples were measured at the Royal NIOZ using a comparable, but slightly different setup. This configuration consists of a NWR193UC (New Wave Research) laser, containing a dual volume ablation cell and an ArF Excimer laser (Existar) with deep UV 193 nm wavelength and <4 ns pulse duration, connected to a quadrupole ICP-MS (iCAP-Q, Thermo Scientific). Energy density of the ablation was also set at 1 J/cm$^2$, the ablation spot was 60 μm in diameter and the repetition rate was 6 Hz for the foraminiferal samples. Calibration to the NIST610 standard was identical to that performed at the Utrecht University. Helium was used as a carrier gas with a flow rate of 0.8 L/min for cell gas and 0.3 L/min for cup gas. From the laser chamber to the ICP-MS, the He flow was mixed with ~0.4 L/min nebulizer Ar and 0.0025 mL/min N$_2$. Before measuring the samples, the nebulizer gas, extraction lens, CCT focus lens and torch position were automatically tuned for the highest sensitivity of $^{238}$U, $^{139}$La, $^{59}$Co and low ThO/Th ratios (<1%) by laser ablating NIST SRM 610 glass. Masses monitored included $^{27}$Al, $^{43}$Ca, $^{44}$Ca, $^{24}$Mg, $^{26}$Mg and $^{55}$Mn. Every ablation lasted approximately 100 seconds, of which the first 20 seconds consisted of a background. The cycle length through all masses was 0,12 seconds. Intensity data were integrated, background subtracted, standardized internally to $^{43}$Ca and calibrated against the NIST SRM 610 signal using Thermo Qtergra software and reference values from Jochum et al. (2011), assuming 40% Ca weight for the foraminiferal samples. JCp-1, MACS-3 and an in-house (foraminiferal) calcite standard (NFHS) were used for quality control and measured every 10 foraminiferal samples. The relative standard deviation in element/Ca based on multiple measurements on

the NFHS is comparable to that of other standards (Mezger et al., 2016). Internal reproducibility of the analyses was all better than 10%, based on the three different carbonate standards.

The resulting Mn and Ca concentrations in foraminiferal test carbonate were used to calculate partition coefficients (D) according to the following equation:

$D_{Mn}=(Mn/Ca)_{calcite} / (Mn/Ca)_{pore water}$.

## 3 Results

### 3.1 Bottom water chemistry and pore water profiles

The oxygen The BWO content varied from 112 µmol/l at station 6 to 33 µmol/l measured at station 9 (Table 1). Stations 7 and 8 were also bathed in dysoxic (<45 µmol/l) bottom water. These low BWO contents were reflected in the shallow
oxygen penetration depths (Fig. 3), measuring less than 5 mm at all sites and reaching a minimum of <2 mm at station 6.
Pore water chemistry, including dissolved oxygen, nitrate, ammonium and manganese, was measured on sites 6, 8 and 10 (Fig. 3). Nitrate concentrations always peaked in the bottom waters (approximately 40 µmol/l), implying an influx of nitrate from the overlying water into the sediments. In all cores, nitrate was rapidly depleted within surficial sediments. Only at station 10, a small subsurface peak was noted in nitrate between 2 and 3 cm depth. The decline in pore water oxygen and
nitrate was accompanied with an increase in ammonium, typically reaching close to 100 µmol/l at 20 cm depth. However, in the top 5 cm, where most of the foraminifera were located (Fontanier et al. 2014), ammonium concentrations were always below 30 µmol/l.
A subsurface peak was observed in pore water Mn concentrations at all sites, suggesting that manganese reduction was taking place within the sediments. However, at station 6 the Mn concentrations were generally low and did not exceed 1.4
20  µmol/l (0.5-1 cm depth interval). At station 8 the subsurface peak was somewhat more developed but the concentrations still remained low (~2 µmol/l), between 0.5 and 1.5 cm depth in sediment. At station 10, the subsurface manganese front was much broader, extending from 2 to 12 cm depth with a maximum concentration of 5.0 µmol/l in the 5-6 cm depth interval.

### 3.2 Mn/Ca ratios in single foraminiferal chambers

Mn/Ca ratios were measured in multiple foraminiferal chambers, ranging from chambers F-1 and F-2 (penultimate or pre-
penultimate) to F-12, and to umbo in *Elphidium batialis*. Selected data of single chamber measurements is presented in Fig. 4, showing *E. batialis, Uvigerina akitaensis* and *Bolivina spissa* from stations with highest numbers of specimens measured across a range of chambers. In addition, single specimen measurements, in which 4 or more chambers were measured, are shown in Appendix 1. For none of the species, or for individual profiles shown in Appendix 1, there was any trend in Mn/Ca ratios with chamber number (Pearson correlation where two-tailed significance was always >0.05). The statistical data
analyses were carried out on all data (see all Mn/Ca data in Appendix 2), confirming the absence of a relation between shell size and Mn/Ca. An example of *Nonionellina labradorica* is not shown in Fig. 4 as relatively low number of total

measurements (n=18) were performed on this species and hence statistical tests are not robust. Further, due to test configuration of *Chilostomellina fimbriata*, only the final chamber (F-0) was analyzed and hence a potential size-related effect could not be determined.

Since no trend was present between Mn/Ca and chamber number, all data were combined for interpretation in the following sections.

### 3.3 Mn/Ca ratios in test calcite and foraminiferal microhabitat distribution

Foraminiferal Mn/Ca ratios were generally low in *E. batialis*, ranging from 0.9 to 33.8 µmol/mol (Fig. 5). The highest concentrations were measured in *N. labradorica* (ranging from 23.4 to 277.0 µmol/mol). *E. batialis* also showed least variability in measurements per sample with average standard error of all measurements per station ranging between 0.2-7.4 µmol/mol.

Most of the measurements were performed on specimens collected from surface sediments, however, for *Uvigerina* spp., *B. spissa*, *N. labradorica* and *C. fimriata* specimens were measured also from deeper sediment intervals (Fig. 5). However, no statistical correlations were observed between the depths where foraminifera were found and their Mn/Ca ratios (Table 3). However, statistical test with *N. labradorica* and *C. fimriata* are of limited value due low total number of specimens measured at these sites.

Average living depth of all species analyzed was calculated after Jorissen et al. (1995). The shallowest living depth was noted for *E. batialis* and was generally encountered in the upper cm of the cores (Fig. 6). At site 10 few isolated specimens were found deeper in sediment resulting in an overall average living depth of 1.2 cm. *Uvigerina* spp. (*U. cf. graciliformis* station 6, *U. akitaensis station* 7, 8 and 9) was found slightly deeper with an ALD ranging from 1.0 cm at station 9 to 2.1 cm at station 7. *B. spissa* was consistently found living deeper than *Uvigerina* spp. with an average living depth close to 2 cm. Of the two deep living taxa, the average living depth of *C. fimbriata* was around 5 cm depth whereas *N. labradorica* was centered at 4 cm depth in sediment, except at station 7 where the ALD of these species were 2.9 and 2.3 cm, respectively. Overall, a systematic distribution of foraminiferal microhabitats was observed with shallow infaunal microhabitat represented by *E. batialis*. Intermediate infaunal habitat was occupied by *Uvigerina* spp. and *B. spissa*, and the deep infaunal habitat was resided by *N. labradorica* and *C. fimbriata*.

Systematic changes were noted in Mn/Ca ratios with respect to foraminiferal microhabitat (Fig. 6). Lowest Mn/Ca values were found in the shallow infaunal *E. batialis*, followed by intermediate infaunal species *Uvigerina* spp. At stations 7, 9 and 10, foraminiferal Mn/Ca rations continued to increase with increasing habitat depth or their ALD. However, an exception was noted at station 8, where the highest Mn/Ca was recorded for deep infaunal species *N. labradorica* and not in the deeper living *C. fimbriata*.

Despite the clear pattern between foraminiferal Mn/Ca with respect to microhabitat distribution, the Mn/Ca concentrations from the average living depth do not exactly match the pore water profiles (Fig. 6), although direct comparisons are not possible for stations 7 and 9. At station 8, for example, the peak in the pore water Mn/Ca is found at a depth of

approximately 1 cm, whereas highest foraminiferal Mn/Ca ratios are found in *N. labradorica* with an ALD of 4.0 cm. At station 10, however, where pore water Mn content is clearly increasing with sediment depth, foraminiferal Mn/Ca ratios show a similar trend. At station 6 where pore water Mn content is generally low, foraminiferal Mn/Ca ratios are also low.

Partitioning coefficients of Mn ($D_{Mn}$) for each taxon were calculated for stations 6, 8 and 10 where pore water data was available (Fig. 7, Appendix 2). Calculations were based on pore water Mn concentrations at the ALD of each species, and their average Mn/Ca ratios. The $D_{Mn}$ of *E. batialis* was very low, ranging from 0.02 at station 10 to 0.03 at station 8. The $D_{Mn}$ of *Uvigerina* spp. was slightly higher, ranging from 0.18 to 0.56; and that of *B. spissa* was similar with an average $D_{Mn}$ of 0.36. The deep infaunal taxa generally had higher $D_{Mn}$, with a coefficient for *N. labradorica* of 1.24 and of 1.77 for *C. fimbriata*. However, at station 10 the calculated $D_{Mn}$ for *N. labradorica* was also low (0.18).

**3.4 Foraminiferal Mn/Ca ratios along the study transect**

The Mn/Ca ratios of *Uvigerina* spp. and *B. spissa* increased with water depth (Fig. 8). Both of these trends were statistically robust with Pearson correlation coefficients of 0.43 (*p<0.01;* n=100) and 0.65 (*p<0.01;* n=79) for *Uvigerina* spp. and *B. spissa*, respectively. Average Mn/Ca ratios of *E. batialis* on the other hand declined slightly along the study transect (Pearson correlation coefficient -0.64 p<0.01 n=65), whereas no trends in Mn/Ca with water depth were found for *N. labradorica* or *C. fimbriata*.

Only the Mn/Ca ratios of *B. spissa* correlated significantly with measured BWO content (Pearson correlation coefficient -0.59; p<0,01; n=79). For *Uvigerina* spp. the highest Mn/Ca ratios coincided with the lowest BWO content. For any of the other taxa no systematic, statistically significant trends were observed between BWO content and Mn/Ca.

**4 Discussion**

**4.1 Intrashell variability in Mn/Ca ratios**

Traditionally, Mn/Ca in foraminiferal test carbonate is used to indicate presence of diagenetic Mn oxyhydroxides and Mn carbonates (e.g. Boyle et al. 1983, Barker et al. 2003). However, studies applying techniques such as LA-ICP-MS, allow circumventing surface contamination by a high depth-resolution during the measurement (e.g. Hathorne et al. 2003, Reichart et al. 2003, Koho et al. 2015). In our study all measurements were also conducted with application of LA-ICP-MS, hence all surficial Mn-contaminants were excluded from data. In addition, all specimens analyzed here were stained with rose Bengal, implying that they were alive, or very recently alive, when collected. Due to the nature of the specimens being very recent, presence of any diagenetic coatings is unlikely and Mn/Ca ratios reflect true Mn-incorporation into the shell walls.

Another advantage of LA-ICP-MS is that it allows measurements of individual foraminiferal chambers, providing information on the changes in the elemental composition in relation to foraminiferal ontogeny/growth. In this study no systematic variations were noted in the Mn/Ca ratios and chamber stages of any foraminifera (Fig. 4, Appendix 1). These observations are consistent with work of Dueñas-Bohórquez (2010) who also noted no clear trend in the Mn/Ca ratios with

chamber stages of *Cibicidoides pachyderma*. Therefore, it appears that Mn/Ca ratios in benthic foraminifera are not substantially influenced by ontogenetic processes, which is occasionally reported for other elements e.g. Mg and B (e.g. Raitzsch et al. 2011), but are primarily driven by environmental changes, such as redox conditions, affecting the concentration of Mn in pore waters. This implies that these foraminifera did not consistently calcify different chambers at
different in-sediment depths, with contrasting Mn concentrations. Effectively this rules out systematic ontogenetic migration across oxygen gradients in the benthic foraminiferal species studied here.

**4.2 Mn/Ca ratios as function of microhabitat**

Foraminifera from three microhabitats (shallow-, intermediate- and deep infauna) were included in this study (Fig. 6): *E. batialis* representing the shallow infaunal microhabitat, *Uvigerina* spp. and *B. spissa* representing the intermediate
microhabitat, and *N. labradorica* and *C. fimbriata* representing the deep infaunal microhabitat. At all stations, the lowest Mn/Ca ratios were measured in the shallow dwelling *E. batialis*. In general with deeper microhabitat distribution, the Mn/Ca ratios appeared to increase (Fig. 6). The only exception seemed to be *C. fimbriata* at station 8 with the average Mn/Ca ratio slightly lower than that of the other deep-infaunal taxa *N. labradorica*. These results are in a good agreement with previous studies on foraminiferal Mn/Ca ratios. In the Baltic Sea, for example, Groeneveld and Filipsson (2013) showed that
specimens of the shallow dwelling *Bulimina marginata* were found to contain no or very small amounts of manganese in their carbonate test, where as elevated Mn/Ca ratios were measured in deep infaunal *Globogulimina turgida*. Moreover the results from the West Pacific presented here are in a good agreement with the TROXCHEM[3] model, a conceptual three-dimensional model, linking foraminiferal Mn uptake, bottom water oxygenation and organic flux (Koho et al., 2015). Based on this model under relatively eutrophic condition, where the bottom waters are still oxygenated very low Mn-concentrations
are found in shallow infaunal species. Deeper in the sediment where higher concentrations of aqueous Mn are present in the pore water (station 10 namely), an increase in foraminiferal Mn/Ca is observed.
Pore water Mn profiles and Mn/Ca ratios in foraminifera in combination with their ALD match relatively closely at station 6 and 10. At station 6, pore water Mn concentrations were generally low; hence the Mn/Ca ratio in the *Uvigerina* spp. was also low. At station 10 where the greatest increase in the pore water Mn content was noted, the deep infaunal *N. labradorica* also
showed much higher Mn/Ca ratios than shallow dwelling *E. batialis*. At station 8, however, where Mn/Ca ratios peaked at just below 1 cm depth in sediment, the highest Mn/Ca ratios were noted in *N. labradorica* with an ALD of 4.0 cm. The apparent mismatch between the pore water profiles and Mn/Ca ratios in foraminifera from their ALD suggests that foraminifera may not always calcify at their observed ALD. As the foraminiferal ALDs represent the average depth where foraminifera are found, they may be skewed by few individuals recovered from deeper or shallower depth intervals. In
addition, bimodal distributions, which were seen for *B. spissa* at station 7 and *Uvigerina* spp. at stations 7 and 8 (Fontanier et al. 2014), can be considered problematic, in case of ALD calculations. However, this does not explain the discrepancy observed at station 8 as the mode of maximum density and ALD of *N. labradorica* were alike, 4.5 cm and 4.0 cm respectively.

Calculated Mn partitioning coefficients (based on pore water concentrations at ALDs) showed a large range in values, ranging from 0.02 for *E. batialis* to 1.77 for *C. fimbriata*. This could imply large offsets between ALD and calcification depths, but partition coefficients for Mn and other elements (e.g. Mg) also have been shown to vary between species (e.g. Toyofuku et al. 2011, Wit et al. 2012, Koho et al. 2015). Previous field-based estimates suggest that the $D_{Mn}$ for benthic foraminifera is generally close to 1 (Glock et al. 2012, Koho et al. 2015), whereas controlled growth experiments of Munsel et al. (2010) estimated that Mn/Ca ratios could be even above 1, with 2.6-10 times higher ratios than in seawater. Irrespective of the observed differences in $D_{Mn}$-values of the species coming from similar depth habitats, it seems that deep infaunal foraminifera, based on their Mn incorporation, are calcifying in or close to the pore waters where they were collected from. The shallow and intermediate infaunal species having $D_{Mn}$ <1 based on their calculated ALD, might calcify somewhat shallower depth where Mn concentrations are lower, or really have substantially lower D-values.

Foraminifera are known migrate in the sediment and laboratory experiments have shown that changes in the sediment oxygenation, typically result in migration of foraminifera to their preferred microhabitat (Gross 2000, Geslin et. al. 2004). Although, no systematic ontogenetic migration was seen in this study, foraminiferal migration is anticipated as all of the studied taxa were found in relatively wider range of sediment depths (Fontanier et al. 2014). Therefore, foraminiferal migration may explain some of the discrepancies seen in the foraminiferal $D_{Mn}$-values, and explain some of the discrepancies between the foraminiferal Mn/Ca ratios and Mn pore water concentrations. Even relatively small scale migration of intermediate and deep infaunal taxa could results in relatively large changes in the ambient pore water Mn content, which would be reflected in their test chemistry during calcification. Furthermore, a closer observation of Mn/Ca ratios of in-sediment dwelling foraminifera shows that both intermediate and deep infaunal species showed relatively higher range of Mn/Ca values at each station (Fig. 4, 6). In contrast, the shallow living *E. batialis*, which is expected to mainly inhabit the surficial more oxygenated sediments, displayed relatively low variability. This suggests that the deeper habitat depth exposes foraminifera to greater variations in pore water Mn concentration. Alternatively, it should be noted that the foraminiferal Mn/Ca measurements were carried out in range of chambers, including younger and older ones (Fig. 4), where as pore water profiles represent a snap shot in time. Therefore, some mismatch can be expected to results from variation in pore water conditions through time.

The relatively high Mn measured in deep-infaunal foraminifera, and for *B. spissa* at station 9 only, further implies that these taxa are actively growing in dysoxic sediments where pore water Mn-concentrations are higher. Although not shown for the species studied here, foraminifera are known to be capable of denitrification (e.g. Risgaard-Petersen et al. 2006, Piña-Ochoa et al. 2010a) and prolonged survival under anoxic conditions (Piña-Ochoa et al. 2010b). Therefore, it is very likely that also the deep-infaunal taxa studied here have adapted similar life strategies. Foraminiferal calcification in the absence of oxygen was also recently demonstrated by Nardelli et al. (2014), whose experimental approach demonstrated that three benthic foraminiferal species *Ammonia tepida*, *Bulimina marginata* and *Cassidulina laevigata* were not only able to survive under anoxic conditions but also form new chambers. Here we show that Mn/Ca ratios in benthic foraminifera can also be measured to identify calcification under such conditions.

**4.3 Implications for paleoceanographic reconstructions**

In recent years, efforts have been made to develop new bottom water oxygenation proxies via application of foraminiferal Mn/Ca ratios (Glock et al. 2012, Groeneveld and Filipsson 2013, McKay et al. 2015, Koho et al. 2015). To date, direct statistically significant correlations between bottom water oxygenation and foraminiferal Mn/Ca ratios have been noted only for the intermediate to deep infaunal *M. barleeanus* (Koho et al. 2015). In this study a statistically significant correlation between Mn/Ca ratio and BWO was also measured in the intermediate infaunal foraminifera *B. spissa* (Pearson correlation coefficient: -0.59, $p<0.01$, n=79). Similarly in the study of Glock et al. (2012) Mn/Ca ratios measured in *B. spissa* from the Peruvian margin seemed to respond to BWO and associated changes in Mn-redox chemistry, although the observed trend was not statistically significant. However, in the case of the other intermediate infaunal species studied here, namely *Uvigerina* spp., no robust statistical correlation with BWO was observed, although the highest Mn/Ca ratios still coincided with the lowest BWO content (33 μmol/l). Consistent with this observation, the highest Mn/Ca ratios in *Uvigerina peregrina* from the Arabian Sea were measured at sites with BWO contents of 20-40 μmol/l (Koho et al. 2015). These observations give further confidence that intermediate infaunal species may be most suitable proxies for BWO and redox reconstructions in the productivity regimes studied here and the study of Koho et al., 2015. Their suitability is most likely related to the vicinity of their microhabitat to the zone of Mn reduction, leading to higher sensitivity for recording changes in redox conditions.

On contrary to intermediate infauna, no clear trends were observed along the study transect in the Mn/Ca ratios of deep or shallow infaunal species (Fig 8). The Mn/Ca ratios were relatively, constantly low in shallow infaunal *E. batialis* or relatively high in deep infaunal species. In the case of the shallow infauna, the surficial microhabitat does not seem to expose foraminifera to pore water Mn, leading to hampering of the any redox signal. Therefore, our data suggests that shallow infaunal taxa may not be suitable for reconstruction of past redox conditions, in line with the results presented in Koho et al (2015). However, the exact response of deep versus intermediate infauna to changes in bottom water oxygenation is most likely to depend on the intricate interplay with organic matter loading. Although productivity at our study site is anticipated to be relatively lower (annual average around 46 mmol/C/m$^2$/d, Yokouchi et al. 2007) than in the Northern Arabian Sea (annual average 111 mmol/C/m$^2$/d, Barber et al. 2001), where the TROXCHEM$^3$ model was developed, fluxes must still be relatively high, as shown by shallow nitrate penetration depth and relatively high ammonium content in the pore waters (Fig. 3). Therefore, influence on the intermediate infauna may also be anticipated here. However, if the carbon loading would be lower, and subsequently Mn-reduction would occur deeper in the sediment, an influence on deeper infauna may be more significant. In paleostudies where large changes in the carbon fluxes are foreseen, Mn/Ca ratios in multiple species, included both intermediate and deep infaunal taxa, should be measured simultaneously.

Along the study transect, the total pore water Mn-inventory did not correlate with BWO content (Fig. 3), having a direct implications for paleoceanographic studies aiming to combine the two. In addition to sedimentary redox chemistry, the total potential pool of Mn in the pore water is related to availability of Mn-oxides in the sedimentary environments (e.g. Van der

Weijden et al. 1999, Law et al. 2009). In this study both intermediate infaunal taxa (*B. spissa* and *Uvigerina* spp.) showed consistent variability in their Mn/Ca ratios along the study gradient (Fig. 8), with ratios increasing with water depth. Therefore, these trends are likely to reflect an increase in the pore water Mn along our study transects as also shown by the pore water Mn profiles (Fig. 3). Concentrations of $Mn^{2+}$ were generally low at station 6, located at around 500 m water depth, where the maximum dissolved $Mn^{2+}$ concentrations were only 1.4 μmol/l at sediment depths of 0.75 cm. With increasing water depth, both, the total depth of the in-sediment zone containing elevated dissolved $Mn^{2+}$ and total concentrations of dissolved manganese increased. At station 10, at a water depth of 2000 m, relatively elevated pore water manganese concentrations were found at sediment depths between 2 and 10 cm with a maximum of 5.0 μmol/l, occurring at the sediment depth of 5.5 cm. This pore water $[Mn^{2+}]$ increase, which is also reflected in the foraminiferal Mn/Ca, may in addition to bottom water oxygenation also be related to an increase in Mn-oxides in the sediment with increasing water depth. Such changes could be due to sustained Mn-recycling, which with no Mn escaping to the water column over time results in the accumulation of high Mn oxides close to the sediment water interface. Alternatively, manganese "shuttling", or downslope transport and focusing of Mn-oxides, is well described in literature (e.g. Schulz et al. 2013, Jilbert et al. 2013), typically explaining spatial differences in the distribution of solid phase manganese along BWO gradients. The slightly higher BWO conditions in the deeper station might have allowed Mn being shuttled there. In addition, although BWO content was higher than 33 μmol/L at all sites during our expedition, at some sites manganese may be able to escape the sedimentary environment at times, leading to relatively Mn depleted pore waters. This may be the case especially at the station 6 and 8, where the Mn-reduction was taking place very close to the sediment surface at the time (Fig 3). Moreover, kinetics of manganese oxidation are known to be relatively slow and subsequently in some aqueous settings $Mn^{2+}$ has been observed to penetrate to the overlying oxic water column in metastable form (e.g. Balzer 1982, Pakhomova et al. 2007), resulting in $Mn^{2+}$ escaping the sedimentary system and diagenetic recycling. At station 10, where the highest pore water $[Mn^{2+}]$ values are noted, Mn oxide reduction is occurring well within the sediment (at depths between 5 and 7 cm). Thus here the internal cycling of Mn-(hydr)oxides is likely to be more efficient, resulting in $Mn^{2+}$ being efficiently trapped within the system (Van der Weijden 1999, Law et al. 2009). Paleoceanographic reconstructions applying Mn/Ca ratios as a proxy for changes in redox chemistry therefore need to take into account changes in availability of Mn-oxides, which could influence sediment biogeochemistry and incorporation of Mn into foraminiferal test carbonate.

## 5 Conclusions

Here we show that Mn/Ca ratios in benthic foraminifera reflect their microhabitat distribution, Mn/Ca ratios increasing with deeper in-sediment habitat. Although appreciable differences between species in Mn partitioning were present, the overall higher Mn/Ca measured in some intermediate and deep-infaunal foraminifera suggest that these taxa are actively growing and calcifying in dysoxic-anoxic sediments where pore water Mn-concentrations are also higher. We also show that Mn incorporation into foraminiferal carbonate appears to reflect the ambient environmental conditions and is not influenced by ontogenetic processes. With regards to paleoceanographic reconstructions, the application of Mn/Ca ratios in intermediate

infaunal foraminifera, such as *B. spissa,* which showed a statistically significant correlation between BWO and Mn/Ca, seems most promising, as their microhabitat appears to expose them to systematic and broad variations in pore water manganese in response to environmental changes.

## 6 Acknowledgements

Authors would like to thank the captain and the crew of R/V Tansei Maru, as well as other cruise participants, for execution of a successful research cruise. Hisami Suga is thanked for taking the bullet train and travelling from Tokyo to Hachinohe port just to supply correct vials and caps for pore water analyses. NWO-ALW (Earth and Life Sciences council) is acknowledge for funding open competition research proposal "Trace metal incorporation in benthic foraminifera: linking ecology and pore water geochemistry" (grant numbers 820.01.011). This work was also supported by Academy of Finland (Project number: 278827, 283453), JSPS KAKENHI (Grant Number 25247085) and the Gravitation grant NESSC from the Dutch Ministry of Education, Culture and Science.

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

**Table 1. Station details including latitude, longitude, water depth and bottom water oxygen (BWO) content. In addition sites, where pore water and foraminifera were collected, are indicated.**

| Station | Latitude (N) | Longitude (E) | Depth (m) | BWO (μmol/l) | Foraminifera | Pore water |
|---|---|---|---|---|---|---|
| 6 | 40° 58.891' | 141° 47.572' | 496 | 112 | Yes | Yes |
| 7 | 41° 10.647' | 141° 47.348' | 760 | 42 | Yes | No |
| 8 | 41° 15.003 | 142° 00.028 | 1033 | 36 | Yes | Yes |
| 9 | 41° 14.982' | 142° 16.969' | 1249 | 33 | Yes | No |
| 10 | 41° 14.918 | 142° 59.989 | 1963 | 70 | Yes | Yes |

**Table 2. Total number of laser ablation measurements and number of foraminifera ablated. In addition, the depth intervals of specimens per station are indicated.**

| Species | Measurements | Specimens | Depth intervals of foraminifera (cm) | | | | |
|---|---|---|---|---|---|---|---|
| | | | ST 6 | ST7 | ST8 | ST9 | ST10 |
| *E. batialis* | 65 | 44 | 0-0.5 | 0-0.5 | 0-0.5 | 0-0.5 | 0-0.5 |
| *Uvigerina* spp. | 100 | 66 | 0-0.5 | 0-0.5 | 0-0.5 | 0-0.5 | 0-0.5; 0.5-1; 1-1.5 |
| *B. spissa* | 79 | 23 | | 0-0.5 | 0-0.5 | 0-0.5; 0.5-1 | |
| *N. labradorica* | 18 | 18 | | 0-0.5 | 0-0.5; 2-2.5; 4-5 | | 0-0.5 |
| *C. fimbriata* | 15 | 15 | | 0-0.5 | 4-5 | 1-1.5; 3-3.5 | |

**Table 3. Pearson correlation coefficients and significance values for Mn/Ca ratios of *Uvigerina* spp. (St 9), *B. spissa* (St 9). *N. labradorica* (St 8) and *C. fimbriata* (St 9) versus sediment depth from where foraminifera were collected from.**

| Species | Pearson correlation | Significance | N |
|---|---|---|---|
| *Uvigerina* spp. | 0.267 | 0.91 | 41 |
| *B. spissa* | 0.017 | 0.937 | 23 |
| *N. labradorica* | 0.512 | 0.159 | 9 |
| *C. fimbriata* | 0.404 | 0.247 | 10 |

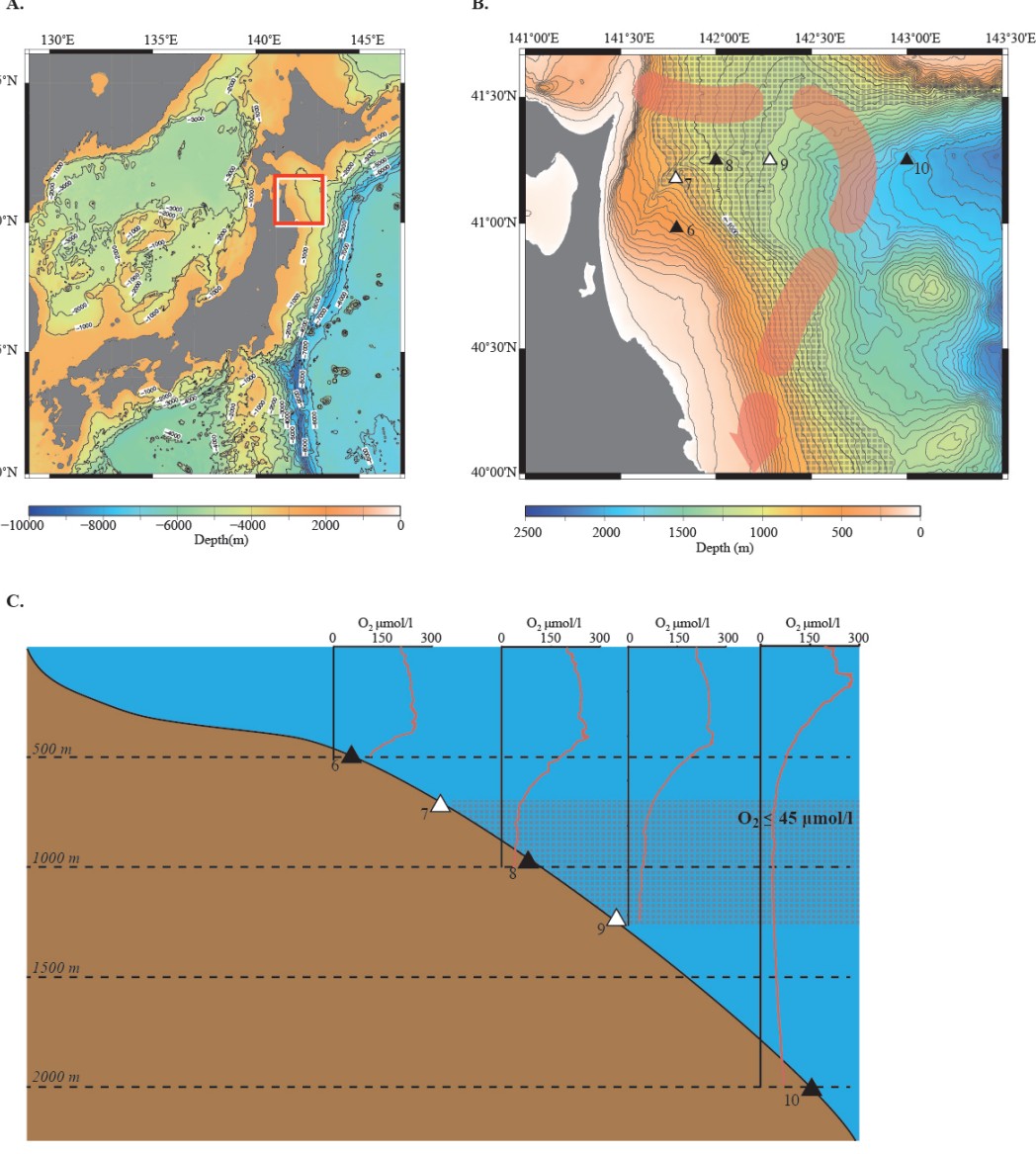

**Figure 1: A** Regional map of the study area **B:** Bathymetric map of the study region, showing the position of Tsugaru warm current (Oguma et al., 2002) and multicore sampling sites. **C:** Schematized study transect with water column profiles of dissolved oxygen. The dysoxic water column (O2<45 µmol/l) is indicated with gray-square pattern.

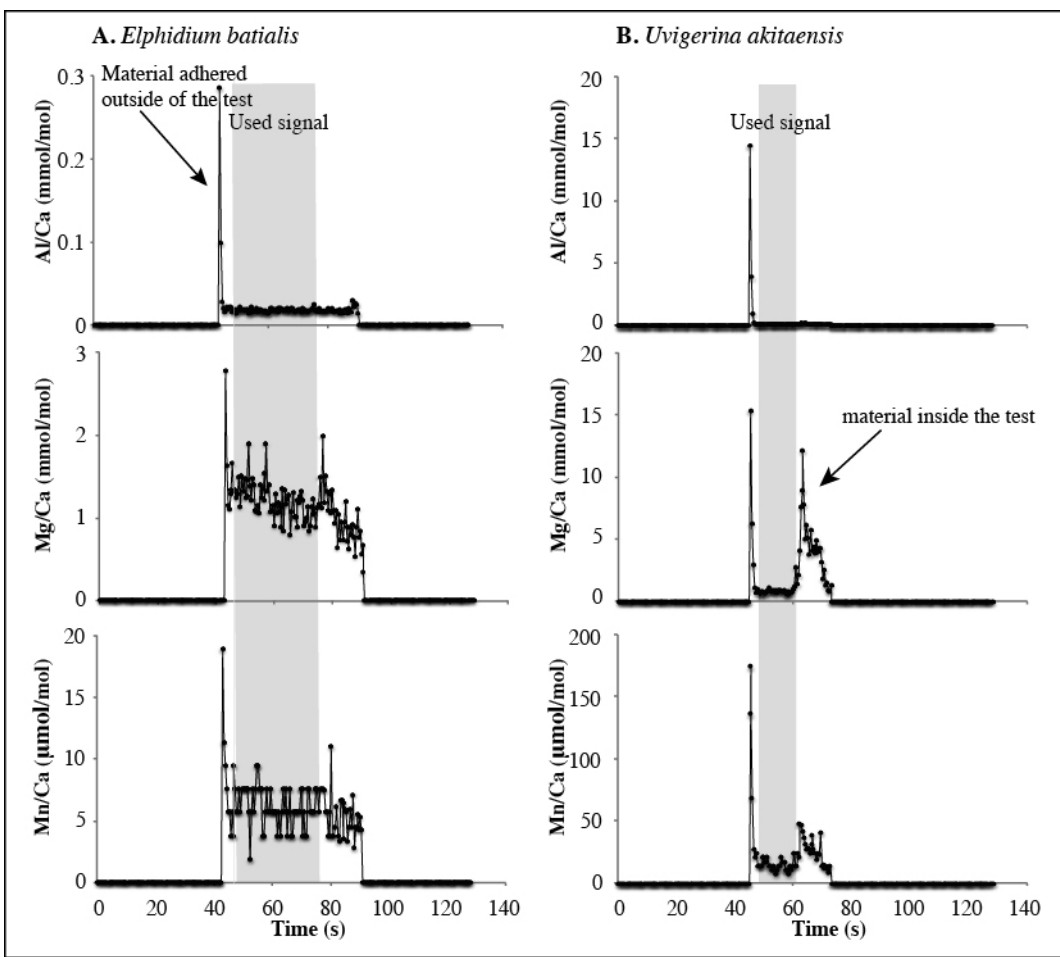

**Figure 2: Laser ablation profile for Al/Ca Mg/Ca and Mn/Ca measured in (A)** *E. baliatils* **(station 8, 0-0.5 cm depth) and (B)** *Uvigerina akitaensis* **(station 7, 0-0.5 cm depth) benthic foraminifera The selected signal for the elemental composition is indicated with the gray shading. Parts of the profile with elevated surface ratios, especially Al/Ca, are removed. In addition, the elevated concentrations, following the ablation through the foraminiferal test are not included in the averaged elemental ratios. Note the different scale bars for elemental ratios.**

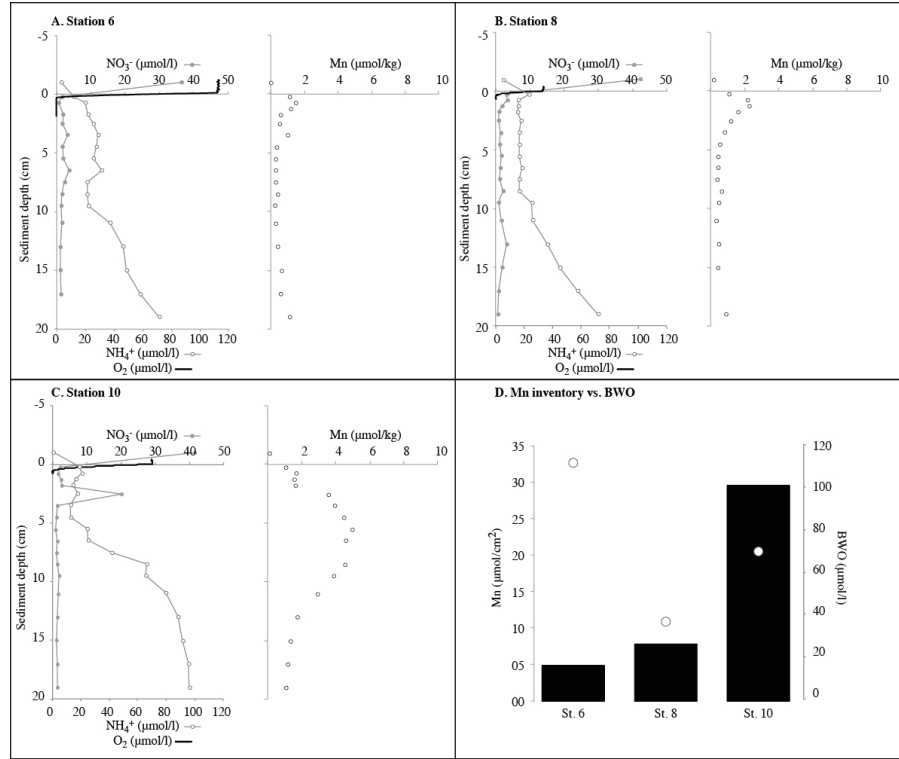

5    **Figure 3: Pore water profiles of dissolved oxygen, nitrate, ammonium and manganese at station 6 (A), 8 (B) and 10 (C). (D) Pore water manganese inventory in the top 10 cm of sediment and bottom water oxygen content (white symbols).**

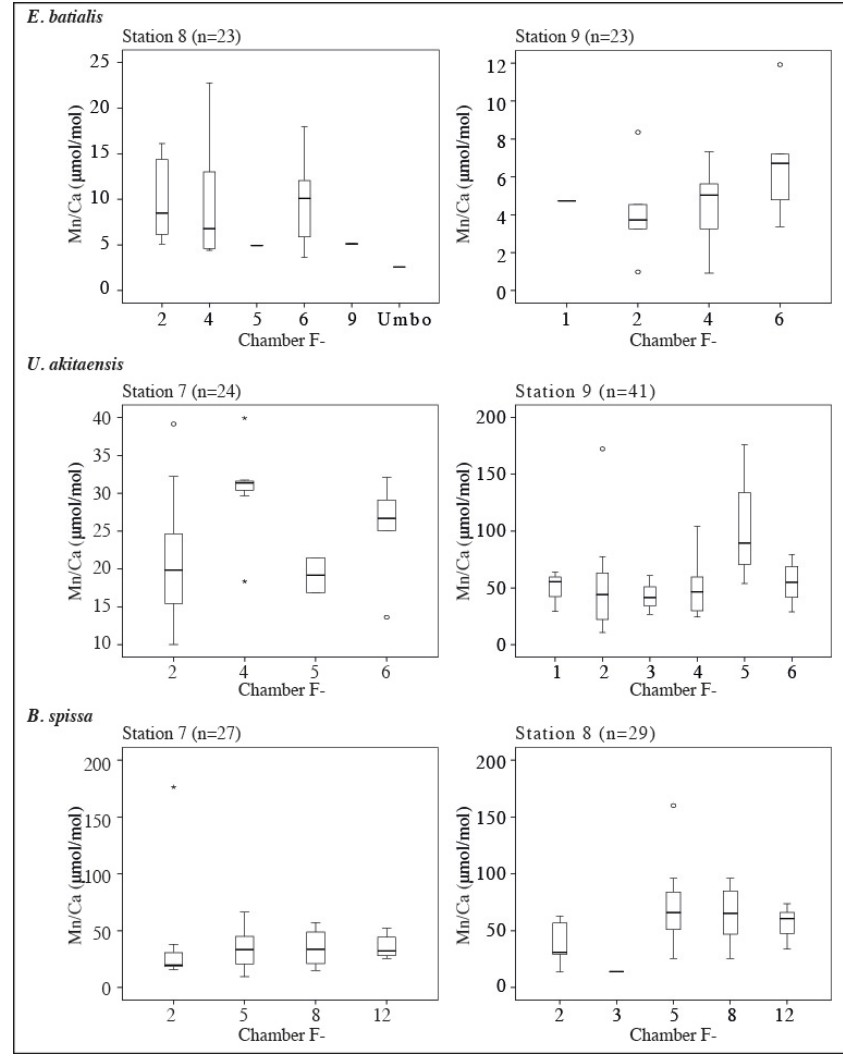

**Figure 4: Box-plots showing chamber-to-chamber variability of Mn/Ca. Error bars display the full range of data variation (from minimum to maximum). Data outliers are represented with an astrix.**

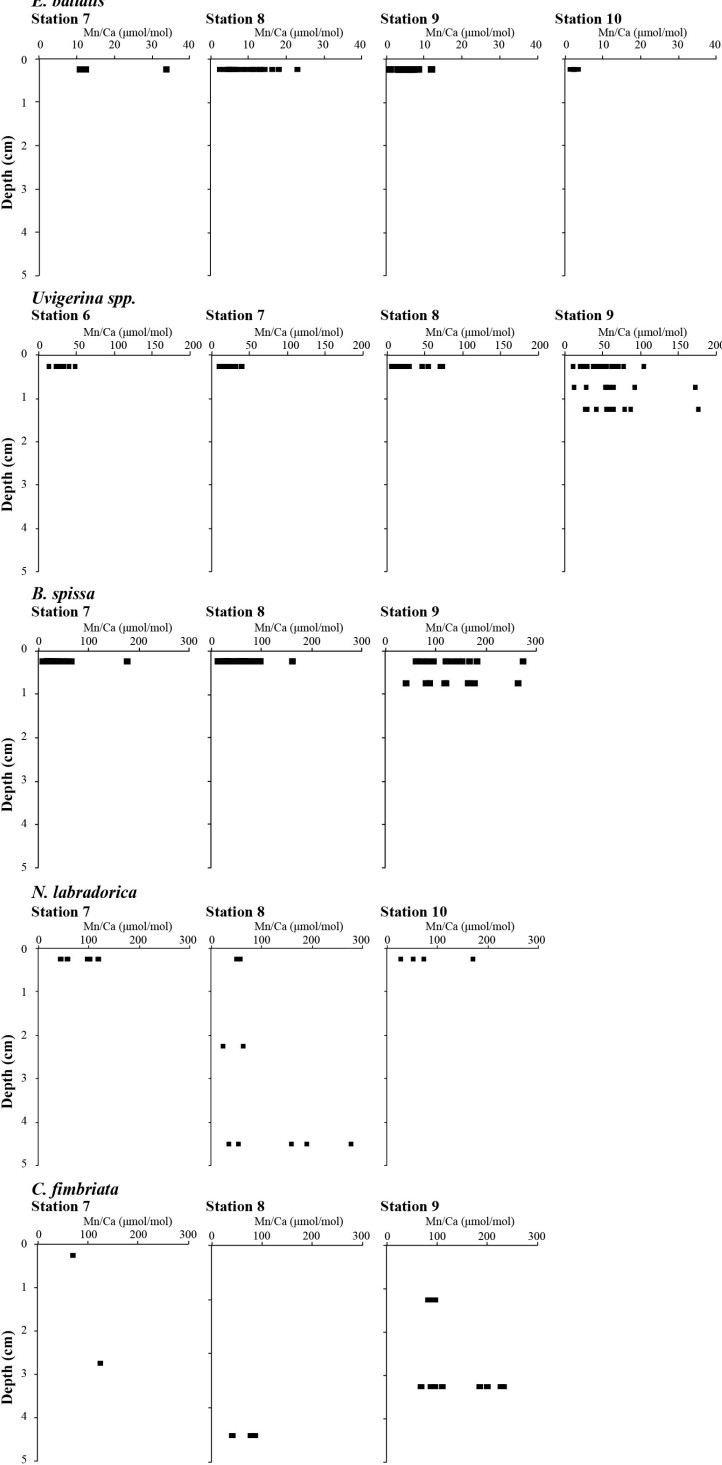

**Figure 5: Individual laser ablation measurements of Mn/Ca in foraminifera versus sediment depth where the specimens were collected.**

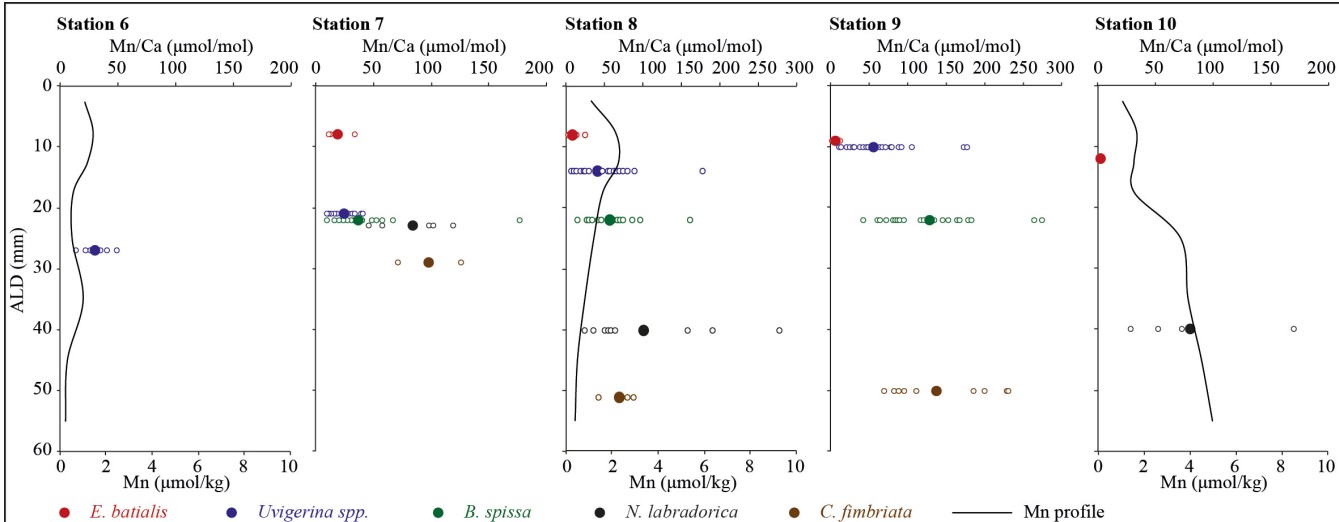

**Figure 6: Mn/Ca ratios in foraminifera as a function of the average living depth of each species. The average of all measurements is indicated with a solid symbol and the individual measurements with open symbols. In addition the pore water profile of Mn is shown in all sites where it was measured.**

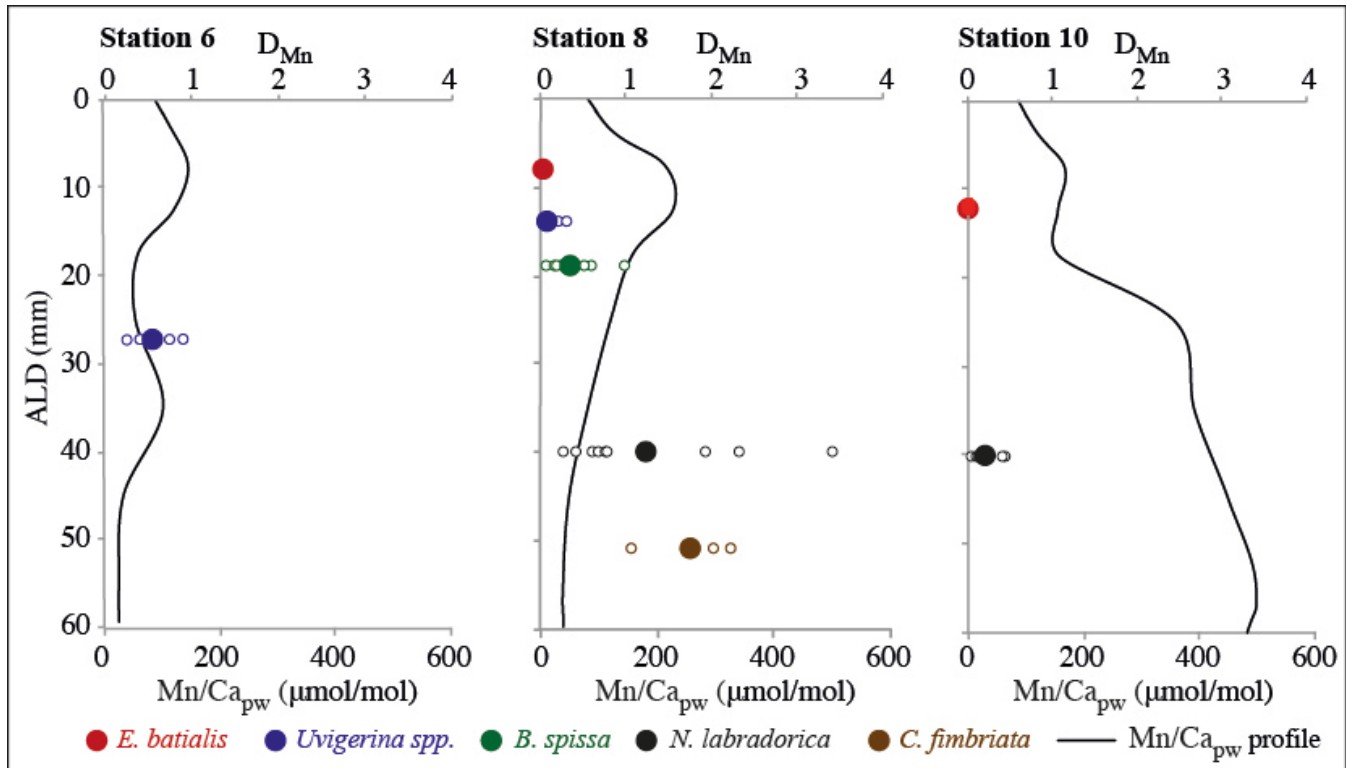

**Figure 7: Manganese partition coefficient DMn in foraminifera as a function of average living depth of each species. In addition, the pore water (pw) Mn/Ca profile is shown.**

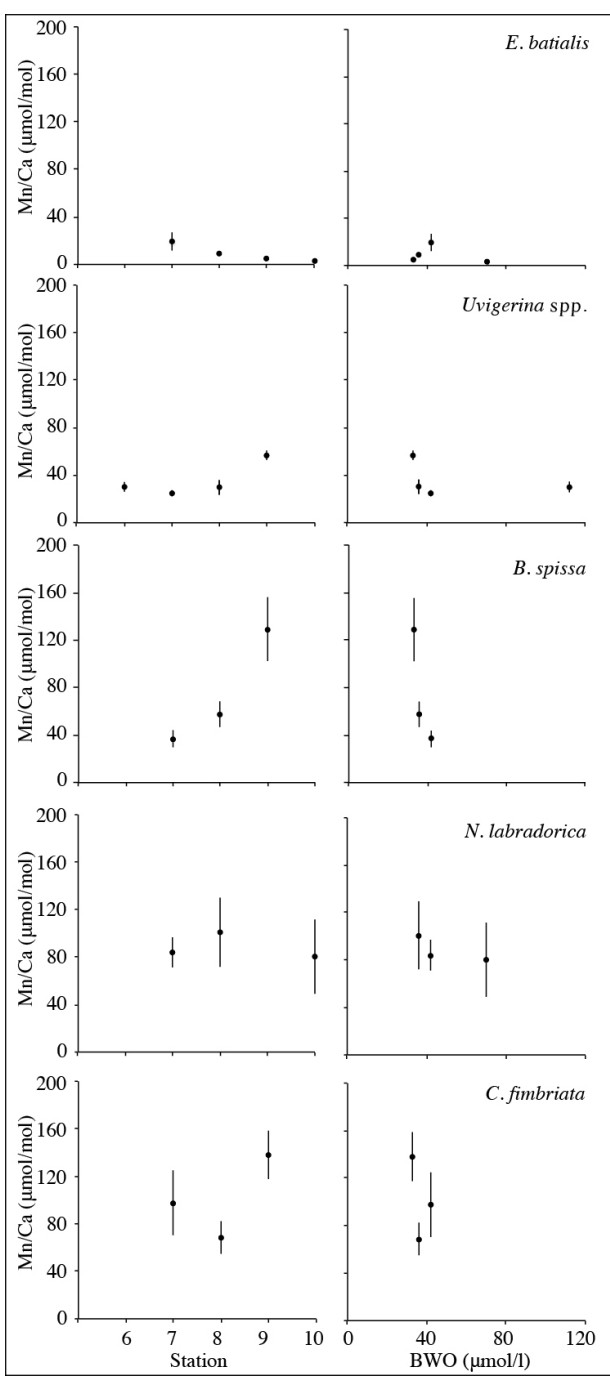

**Figure 8: Variability of average Mn/Ca ratios of each species plotted against the study transect from station 6 to station 10 (left), and along the bottom water oxygenation (right). The error bars represent the standard error of the measurements.**

## Appendix I. Mn/Ca ratios in individual foraminiferal chambers.

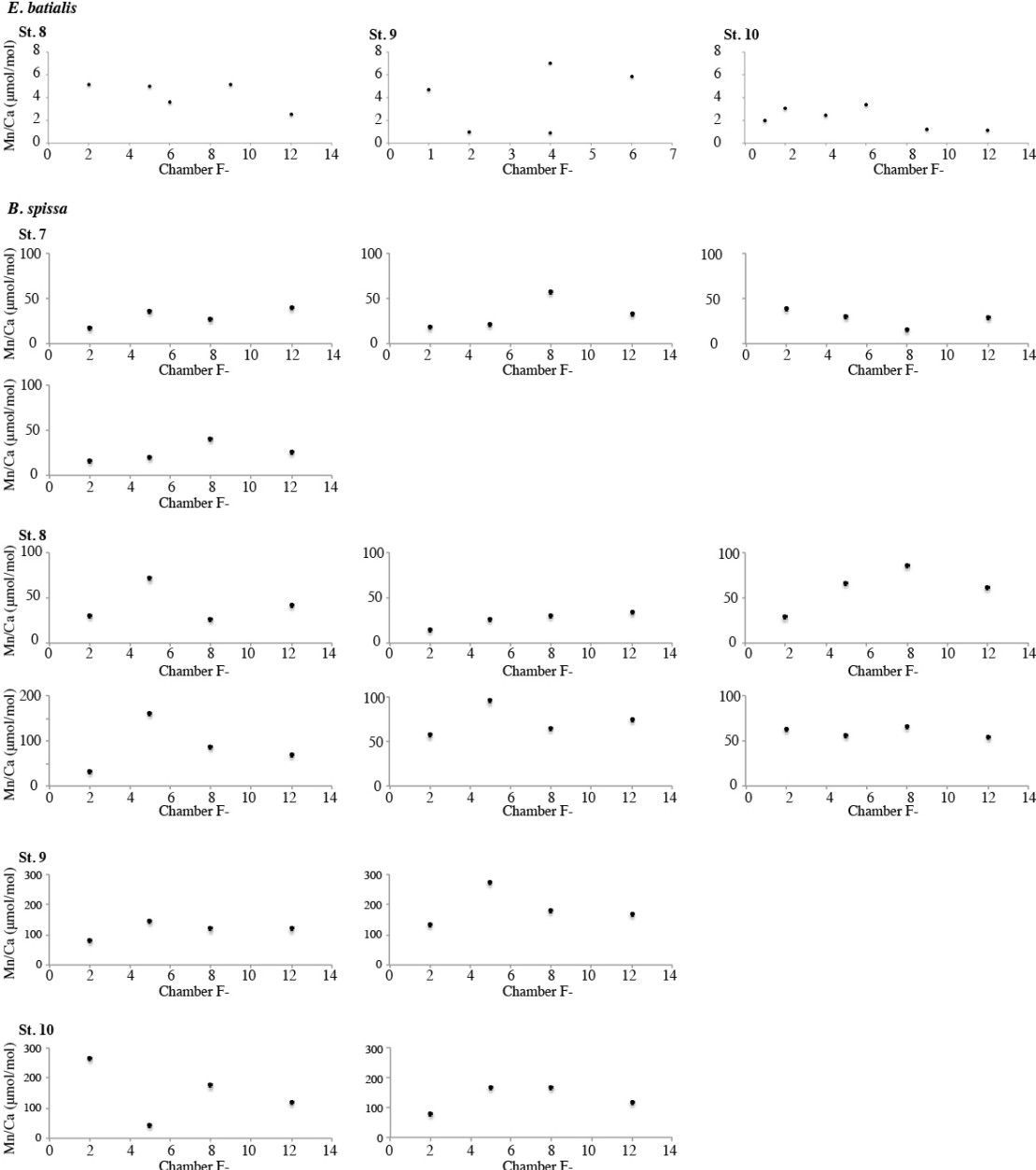

APPENDIX 2: Raw data

| Specimen id | Species | Station | Chamber F= | Sed depth (cm) | Mn/Ca umol/mol | Average living depth cm | Mn/Ca$_{pw}$ at ALD umol/mol | DMn | Dmn average per station |
|---|---|---|---|---|---|---|---|---|---|
| 30 | E. batialis | 7 | 4 | 0-0,5 | 12,58 | 0,4 | no data | | |
| 30 | E. batialis | 7 | 2 | 0.-0.5 | 33,78 | 0,4 | no data | | |
| 30 | E. batialis | 7 | 6 | 0.-0.5 | 10,85 | 0,4 | no data | | |
| 31 | E. batialis | 8 | 4 | 0-0,5 | 4,41 | 0,8 | 212,3 | 0,02 | 0,04 |
| 31 | E. batialis | 8 | 2 | 0.-0.5 | 14,36 | 0,8 | 212,3 | 0,07 | |
| 31 | E. batialis | 8 | 6 | 0.-0.5 | 9,39 | 0,8 | 212,3 | 0,04 | |
| 32 | E. batialis | 8 | 4 | 0-0,5 | 22,78 | 0,8 | 212,3 | 0,11 | |
| 32 | E. batialis | 8 | 6 | 0.-0.5 | 6,97 | 0,8 | 212,3 | 0,03 | |
| 33 | E. batialis | 8 | 4 | 0-0,5 | 13,00 | 0,8 | 212,3 | 0,06 | |
| 33 | E. batialis | 8 | 6 | 0.-0.5 | 17,98 | 0,8 | 212,3 | 0,08 | |
| 34 | E. batialis | 8 | 4 | 0-0,5 | 6,77 | 0,8 | 212,3 | 0,03 | |
| 34 | E. batialis | 8 | 2 | 0.-0.5 | 8,48 | 0,8 | 212,3 | 0,04 | |
| 34 | E. batialis | 8 | 6 | 0.-0.5 | 12,96 | 0,8 | 212,3 | 0,06 | |
| 35 | E. batialis | 8 | 5 | 0-0,5 | 4,93 | 0,8 | 212,3 | 0,02 | |
| 35 | E. batialis | 8 | 9 | 0-0,5 | 5,12 | 0,8 | 212,3 | 0,02 | |
| 35 | E. batialis | 8 | 12 | 0-0,5 | 2,56 | 0,8 | 212,3 | 0,01 | |
| 35 | E. batialis | 8 | 2 | 0.-0.5 | 5,10 | 0,8 | 212,3 | 0,02 | |
| 35 | E. batialis | 8 | 6 | 0.-0.5 | 3,62 | 0,8 | 212,3 | 0,02 | |
| 36 | E. batialis | 8 | 4 | 0-0,5 | 4,62 | 0,8 | 212,3 | 0,02 | |
| 36 | E. batialis | 8 | 2 | 0.-0.5 | 6,17 | 0,8 | 212,3 | 0,03 | |
| 36 | E. batialis | 8 | 6 | 0.-0.5 | 5,88 | 0,8 | 212,3 | 0,03 | |
| 37 | E. batialis | 8 | 6 | 0.-0.5 | 10,96 | 0,8 | 212,3 | 0,05 | |
| 38 | E. batialis | 8 | 6 | 0.-0.5 | 5,10 | 0,8 | 212,3 | 0,02 | |
| 39 | E. batialis | 8 | 6 | 0.-0.5 | 10,79 | 0,8 | 212,3 | 0,05 | |
| 40 | E. batialis | 8 | 2 | 0.-0.5 | 16,13 | 0,8 | 212,3 | 0,08 | |
| 40 | E. batialis | 8 | 6 | 0.-0.5 | 12,07 | 0,8 | 212,3 | 0,06 | |
| 41 | E. batialis | 9 | 4 | 0-0,5 | 3,10 | 0,9 | no data | | |
| 41 | E. batialis | 9 | 2 | 0.-0.5 | 3,26 | 0,9 | no data | | |
| 41 | E. batialis | 9 | 6 | 0.-0.5 | 3,37 | 0,9 | no data | | |
| 42 | E. batialis | 9 | 4 | 0-0,5 | 5,04 | 0,9 | no data | | |
| 42 | E. batialis | 9 | 6 | 0.-0.5 | 6,59 | 0,9 | no data | | |
| 43 | E. batialis | 9 | 4 | 0-0,5 | 3,25 | 0,9 | no data | | |
| 43 | E. batialis | 9 | 2 | 0.-0.5 | 8,35 | 0,9 | no data | | |
| 43 | E. batialis | 9 | 6 | 0.-0.5 | 7,23 | 0,9 | no data | | |
| 44 | E. batialis | 9 | 4 | 0-0,5 | 4,93 | 0,9 | no data | | |
| 44 | E. batialis | 9 | 2 | 0.-0.5 | 4,53 | 0,9 | no data | | |
| 44 | E. batialis | 9 | 6 | 0.-0.5 | 6,84 | 0,9 | no data | | |
| 45 | E. batialis | 9 | 4 | 0-0,5 | 5,59 | 0,9 | no data | | |
| 45 | E. batialis | 9 | 2 | 0.-0.5 | 3,73 | 0,9 | no data | | |
| 45 | E. batialis | 9 | 6 | 0.-0.5 | 3,71 | 0,9 | no data | | |
| 46 | E. batialis | 9 | 4 | 0-0,5 | 7,32 | 0,9 | no data | | |
| 46 | E. batialis | 9 | 6 | 0.-0.5 | 11,92 | 0,9 | no data | | |
| 47 | E. batialis | 9 | 4 | 0-0,5 | 5,63 | 0,9 | no data | | |
| 47 | E. batialis | 9 | 6 | 0.-0.5 | 7,15 | 0,9 | no data | | |
| 48 | E. batialis | 9 | 4 | 0-0,5 | 0,92 | 0,9 | no data | | |
| 48 | E. batialis | 9 | 1 | 0-0,5 | 4,74 | 0,9 | no data | | |
| 48 | E. batialis | 9 | 4 | 0-0,5 | 7,08 | 0,9 | no data | | |
| 48 | E. batialis | 9 | 2 | 0.-0.5 | 0,98 | 0,9 | no data | | |
| 48 | E. batialis | 9 | 6 | 0.-0.5 | 5,86 | 0,9 | no data | | |
| 49 | E. batialis | 10 | 9 | 0-0,5 | 1,24 | 1,2 | 154,8 | 0,01 | 0,02 |
| 49 | E. batialis | 10 | 1 | 0-0,5 | 2,00 | 1,2 | 154,8 | 0,01 | |
| 49 | E. batialis | 10 | 4 | 0-0,5 | 2,51 | 1,2 | 154,8 | 0,02 | |
| 49 | E. batialis | 10 | 12 | 0-0,5 | 1,13 | 1,2 | 154,8 | 0,01 | |
| 49 | E. batialis | 10 | 2 | 0.-0.5 | 3,06 | 1,2 | 154,8 | 0,02 | |
| 49 | E. batialis | 10 | 6 | 0.-0.5 | 3,39 | 1,2 | 154,8 | 0,02 | |
| 50 | E. batialis | 10 | 4 | 0-0,5 | 2,22 | 1,2 | 154,8 | 0,01 | |
| 50 | E. batialis | 10 | 2 | 0.-0.5 | 1,51 | 1,2 | 154,8 | 0,01 | |
| 50 | E. batialis | 10 | 6 | 0.-0.5 | 2,18 | 1,2 | 154,8 | 0,01 | |
| 51 | E. batialis | 10 | 4 | 0-0,5 | 3,57 | 1,2 | 154,8 | 0,02 | |
| 51 | E. batialis | 10 | 6 | 0.-0.5 | 3,06 | 1,2 | 154,8 | 0,02 | |
| 52 | E. batialis | 10 | 4 | 0-0,5 | 3,41 | 1,2 | 154,8 | 0,02 | |
| 52 | E. batialis | 10 | 2 | 0.-0.5 | 2,33 | 1,2 | 154,8 | 0,02 | |
| 52 | E. batialis | 10 | 6 | 0.-0.5 | 3,49 | 1,2 | 154,8 | 0,02 | |
| 53 | E. batialis | 10 | 4 | 0-0,5 | 1,79 | 1,2 | 154,8 | 0,01 | |
| 53 | E. batialis | 10 | 6 | 0.-0.5 | 2,31 | 1,2 | 154,8 | 0,01 | |

| Specimen id | Species | Station | Chamber F= | Sed depth (cm) | Mn/Ca umol/mol | Average living depth cm | Mn/Ca$_{pw}$ at ALD umol/mol | DMn | DMn average per station |
|---|---|---|---|---|---|---|---|---|---|
| 44 | B. spissa | 7 | 2 | 0-0,5 | 17,83 | 2,2 | no data | | |
| 44 | B. spissa | 7 | 5 | 0-0,5 | 36,28 | 2,2 | no data | | |
| 44 | B. spissa | 7 | 8 | 0-0,5 | 27,00 | 2,2 | no data | | |
| 44 | B. spissa | 7 | 12 | 0-0,5 | 39,70 | 2,2 | no data | | |
| 45 | B. spissa | 7 | 2 | 0-0,5 | 19,23 | 2,2 | no data | | |
| 45 | B. spissa | 7 | 5 | 0-0,5 | 22,01 | 2,2 | no data | | |
| 45 | B. spissa | 7 | 8 | 0-0,5 | 57,29 | 2,2 | no data | | |
| 45 | B. spissa | 7 | 12 | 0-0,5 | 32,21 | 2,2 | no data | | |
| 46 | B. spissa | 7 | 2 | 0-0,5 | 38,09 | 2,2 | no data | | |
| 46 | B. spissa | 7 | 5 | 0-0,5 | 30,47 | 2,2 | no data | | |
| 46 | B. spissa | 7 | 8 | 0-0,5 | 15,07 | 2,2 | no data | | |
| 46 | B. spissa | 7 | 12 | 0-0,5 | 28,81 | 2,2 | no data | | |
| 47 | B. spissa | 7 | 2 | 0-0,5 | 15,73 | 2,2 | no data | | |
| 47 | B. spissa | 7 | 5 | 0-0,5 | 19,65 | 2,2 | no data | | |
| 47 | B. spissa | 7 | 8 | 0-0,5 | 40,36 | 2,2 | no data | | |
| 47 | B. spissa | 7 | 12 | 0-0,5 | 25,62 | 2,2 | no data | | |
| 48 | B. spissa | 7 | 2 | 0-0,5 | 23,37 | 2,2 | no data | | |
| 48 | B. spissa | 7 | 5 | 0-0,5 | 9,61 | 2,2 | no data | | |
| 49 | B. spissa | 7 | 2 | 0-0,5 | 19,23 | 2,2 | no data | | |
| 49 | B. spissa | 7 | 5 | 0-0,5 | 37,67 | 2,2 | no data | | |
| 49 | B. spissa | 7 | 12 | 0-0,5 | 27,29 | 2,2 | no data | | |
| 50 | B. spissa | 7 | 2 | 0-0,5 | 176,40 | 2,2 | no data | | |
| 50 | B. spissa | 7 | 5 | 0-0,5 | 52,07 | 2,2 | no data | | |
| 50 | B. spissa | 7 | 12 | 0-0,5 | 48,70 | 2,2 | no data | | |
| 51 | B. spissa | 7 | 2 | 0-0,5 | 19,99 | 2,2 | no data | | |
| 51 | B. spissa | 7 | 5 | 0-0,5 | 66,24 | 2,2 | no data | | |
| 51 | B. spissa | 7 | 12 | 0-0,5 | 52,07 | 2,2 | no data | | |
| 52 | B. spissa | 8 | 2 | 0-0,5 | 29,70 | 1,9 | 158,8 | 0,19 | 0,36 |
| 52 | B. spissa | 8 | 5 | 0-0,5 | 72,02 | 1,9 | 158,8 | 0,45 | |
| 52 | B. spissa | 8 | 8 | 0-0,5 | 25,65 | 1,9 | 158,8 | 0,16 | |
| 52 | B. spissa | 8 | 12 | 0-0,5 | 41,17 | 1,9 | 158,8 | 0,26 | |
| 53 | B. spissa | 8 | 2 | 0-0,5 | 29,27 | 1,9 | 158,8 | 0,18 | |
| 53 | B. spissa | 8 | 5 | 0-0,5 | 65,83 | 1,9 | 158,8 | 0,41 | |
| 53 | B. spissa | 8 | 8 | 0-0,5 | 85,07 | 1,9 | 158,8 | 0,54 | |
| 53 | B. spissa | 8 | 12 | 0-0,5 | 61,47 | 1,9 | 158,8 | 0,39 | |
| 54 | B. spissa | 8 | 2 | 0-0,5 | 56,79 | 1,9 | 158,8 | 0,36 | |
| 54 | B. spissa | 8 | 5 | 0-0,5 | 96,07 | 1,9 | 158,8 | 0,60 | |
| 54 | B. spissa | 8 | 8 | 0-0,5 | 64,18 | 1,9 | 158,8 | 0,40 | |
| 54 | B. spissa | 8 | 12 | 0-0,5 | 73,67 | 1,9 | 158,8 | 0,46 | |
| 55 | B. spissa | 8 | 2 | 0-0,5 | 13,82 | 1,9 | 158,8 | 0,09 | |
| 55 | B. spissa | 8 | 5 | 0-0,5 | 25,61 | 1,9 | 158,8 | 0,16 | |
| 55 | B. spissa | 8 | 8 | 0-0,5 | 29,69 | 1,9 | 158,8 | 0,19 | |
| 55 | B. spissa | 8 | 12 | 0-0,5 | 33,77 | 1,9 | 158,8 | 0,21 | |
| 56 | B. spissa | 8 | 2 | 0-0,5 | 32,35 | 1,9 | 158,8 | 0,20 | |
| 56 | B. spissa | 8 | 5 | 0-0,5 | 160,07 | 1,9 | 158,8 | 1,01 | |
| 56 | B. spissa | 8 | 8 | 0-0,5 | 85,01 | 1,9 | 158,8 | 0,54 | |
| 56 | B. spissa | 8 | 12 | 0-0,5 | 68,27 | 1,9 | 158,8 | 0,43 | |
| 57 | B. spissa | 8 | 2 | 0-0,5 | 62,74 | 1,9 | 158,8 | 0,40 | |
| 57 | B. spissa | 8 | 5 | 0-0,5 | 56,12 | 1,9 | 158,8 | 0,35 | |
| 57 | B. spissa | 8 | 8 | 0-0,5 | 65,33 | 1,9 | 158,8 | 0,41 | |
| 57 | B. spissa | 8 | 12 | 0-0,5 | 54,11 | 1,9 | 158,8 | 0,34 | |
| 58 | B. spissa | 8 | 3 | 0-0,5 | 14,04 | 1,9 | 158,8 | 0,09 | |
| 58 | B. spissa | 8 | 8 | 0-0,5 | 96,10 | 1,9 | 158,8 | 0,61 | |
| 58 | B. spissa | 8 | 12 | 0-0,5 | 63,73 | 1,9 | 158,8 | 0,40 | |
| 59 | B. spissa | 8 | 5 | 0-0,5 | 45,63 | 1,9 | 158,8 | 0,29 | |
| 59 | B. spissa | 8 | 12 | 0-0,5 | 59,80 | 1,9 | 158,8 | 0,38 | |
| 60 | B. spissa | 9 | 2 | 0-0,5 | 83,05 | 2,2 | no data | | |
| 60 | B. spissa | 9 | 5 | 0-0,5 | 144,96 | 2,2 | no data | | |
| 60 | B. spissa | 9 | 8 | 0-0,5 | 121,06 | 2,2 | no data | | |
| 60 | B. spissa | 9 | 12 | 0-0,5 | 119,73 | 2,2 | no data | | |
| 61 | B. spissa | 9 | 2 | 0-0,5 | 133,37 | 2,2 | no data | | |
| 61 | B. spissa | 9 | 5 | 0-0,5 | 273,02 | 2,2 | no data | | |
| 61 | B. spissa | 9 | 8 | 0-0,5 | 181,53 | 2,2 | no data | | |
| 61 | B. spissa | 9 | 12 | 0-0,5 | 165,67 | 2,2 | no data | | |
| 63 | B. spissa | 9 | 8 | 0-0,5 | 62,66 | 2,2 | no data | | |
| 63 | B. spissa | 9 | 12 | 0-0,5 | 71,88 | 2,2 | no data | | |
| 64 | B. spissa | 9 | 5 | 0-0,5 | 152,53 | 2,2 | no data | | |
| 64 | B. spissa | 9 | 8 | 0-0,5 | 94,90 | 2,2 | no data | | |
| 64 | B. spissa | 9 | 12 | 0-0,5 | 60,20 | 2,2 | no data | | |
| 65 | B. spissa | 9 | 2 | 0,5-1 | 264,03 | 2,2 | no data | | |
| 65 | B. spissa | 9 | 5 | 0,5-1 | 41,21 | 2,2 | no data | | |
| 65 | B. spissa | 9 | 8 | 0,5-1 | 177,07 | 2,2 | no data | | |
| 65 | B. spissa | 9 | 12 | 0,5-1 | 118,65 | 2,2 | no data | | |
| 67 | B. spissa | 9 | 2 | 0,5-1 | 79,88 | 2,2 | no data | | |
| 67 | B. spissa | 9 | 5 | 0,5-1 | 163,35 | 2,2 | no data | | |
| 67 | B. spissa | 9 | 8 | 0,5-1 | 164,15 | 2,2 | no data | | |
| 67 | B. spissa | 9 | 12 | 0,5-1 | 116,29 | 2,2 | no data | | |
| 68 | B. spissa | 9 | 2 | 0,5-1 | 88,22 | 2,2 | no data | | |
| 68 | B. spissa | 9 | 12 | 0,5-1 | 88,00 | 2,2 | no data | | |

| Specimen id | Species | Station | Chamber F= | Sed depth (cm) | Mn/Ca umol/mol | Average living depth cm | Mn/Ca$_{pw}$ at ALD umol/mol | D$_{Mn}$ | D$_{Mn}$ average per station |
|---|---|---|---|---|---|---|---|---|---|
| 6 | U. cf. graciliformis | 6 | 2 | 0-0,5 | 26,15 | 2,7 | 53,6 | 0,49 | 0,56 |
| 6 | U. cf. graciliformis | 6 | 4 | 0-0,5 | 22,35 | 2,7 | 53,6 | 0,42 | |
| 7 | U. cf. graciliformis | 6 | 1 | 0-0,5 | 13,30 | 2,7 | 53,6 | 0,25 | |
| 7 | U. cf. graciliformis | 6 | 4 | 0-0,5 | 40,03 | 2,7 | 53,6 | 0,75 | |
| 8 | U. cf. graciliformis | 6 | 4 | 0-0,5 | 30,96 | 2,7 | 53,6 | 0,58 | |
| 9 | U. cf. graciliformis | 6 | 1 | 0-0,5 | 25,32 | 2,7 | 53,6 | 0,47 | |
| 9 | U. cf. graciliformis | 6 | 3 | 0-0,5 | 32,69 | 2,7 | 53,6 | 0,61 | |
| 11 | U. cf. graciliformis | 6 | 2 | 0-0,5 | 48,08 | 2,7 | 53,6 | 0,90 | |
| 11 | U. cf. graciliformis | 6 | 4 | 0-0,5 | 32,66 | 2,7 | 53,6 | 0,61 | |
| 12 | U. cf. graciliformis | 6 | 1 | 0-0,5 | 27,36 | 2,7 | 53,6 | 0,51 | |
| 12 | U. cf. graciliformis | 6 | 4 | 0-0,5 | 33,91 | 2,7 | 53,6 | 0,63 | |
| 62 | U. cf. graciliformis | 6 | 1 | 0.-0.5 | 28,40 | 2,7 | 53,6 | 0,53 | |
| 64 | U. akitaensis | 7 | 4 | 0-0,5 | 18,36 | 2,1 | no data | | |
| 64 | U. akitaensis | 7 | 2 | 0.-0.5 | 39,13 | 2,1 | no data | | |
| 65 | U. akitaensis | 7 | 4 | 0-0,5 | 31,72 | 2,1 | no data | | |
| 65 | U. akitaensis | 7 | 2 | 0.-0.5 | 20,62 | 2,1 | no data | | |
| 66 | U. akitaensis | 7 | 4 | 0-0,5 | 29,64 | 2,1 | no data | | |
| 66 | U. akitaensis | 7 | 6 | 0.-0.5 | 13,63 | 2,1 | no data | | |
| 67 | U. akitaensis | 7 | 4 | 0-0,5 | 31,34 | 2,1 | no data | | |
| 67 | U. akitaensis | 7 | 2 | 0.-0.5 | 24,63 | 2,1 | no data | | |
| 67 | U. akitaensis | 7 | 6 | 0.-0.5 | 26,12 | 2,1 | no data | | |
| 68 | U. akitaensis | 7 | 4 | 0-0,5 | 31,19 | 2,1 | no data | | |
| 68 | U. akitaensis | 7 | 2 | 0.-0.5 | 19,86 | 2,1 | no data | | |
| 68 | U. akitaensis | 7 | 6 | 0.-0.5 | 27,27 | 2,1 | no data | | |
| 69 | U. akitaensis | 7 | 4 | 0-0,5 | 31,49 | 2,1 | no data | | |
| 69 | U. akitaensis | 7 | 2 | 0.-0.5 | 32,20 | 2,1 | no data | | |
| 69 | U. akitaensis | 7 | 6 | 0.-0.5 | 29,12 | 2,1 | no data | | |
| 70 | U. akitaensis | 7 | 4 | 0-0,5 | 39,90 | 2,1 | no data | | |
| 70 | U. akitaensis | 7 | 2 | 0.-0.5 | 15,47 | 2,1 | no data | | |
| 70 | U. akitaensis | 7 | 6 | 0.-0.5 | 32,11 | 2,1 | no data | | |
| 71 | U. akitaensis | 7 | 5 | 0.-0.5 | 21,48 | 2,1 | no data | | |
| 71 | U. akitaensis | 7 | 2 | 0.-0.5 | 15,64 | 2,1 | no data | | |
| 72 | U. akitaensis | 7 | 2 | 0.-0.5 | 13,32 | 2,1 | no data | | |
| 72 | U. akitaensis | 7 | 6 | 0.-0.5 | 25,05 | 2,1 | no data | | |
| 73 | U. akitaensis | 7 | 2 | 0.-0.5 | 10,03 | 2,1 | no data | | |
| 73 | U. akitaensis | 7 | 5 | 0.-0.5 | 16,87 | 2,1 | no data | | |
| 74 | U. akitaensis | 8 | 4 | 0-0,5 | 53,69 | 1,4 | 227,4 | 0,24 | 0,13 |
| 74 | U. akitaensis | 8 | 2 | 0.-0.5 | 73,59 | 1,4 | 227,4 | 0,32 | |
| 74 | U. akitaensis | 8 | 5 | 0.-0.5 | 69,33 | 1,4 | 227,4 | 0,30 | |
| 75 | U. akitaensis | 8 | 4 | 0-0,5 | 29,72 | 1,4 | 227,4 | 0,13 | |
| 75 | U. akitaensis | 8 | 6 | 0.-0.5 | 54,70 | 1,4 | 227,4 | 0,24 | |
| 76 | U. akitaensis | 8 | 2 | 0.-0.5 | 26,98 | 1,4 | 227,4 | 0,12 | |
| 76 | U. akitaensis | 8 | 6 | 0.-0.5 | 28,80 | 1,4 | 227,4 | 0,13 | |
| 77 | U. akitaensis | 8 | 4 | 0-0,5 | 9,51 | 1,4 | 227,4 | 0,04 | |
| 77 | U. akitaensis | 8 | 2 | 0.-0.5 | 8,46 | 1,4 | 227,4 | 0,04 | |
| 77 | U. akitaensis | 8 | 6 | 0.-0.5 | 45,80 | 1,4 | 227,4 | 0,20 | |
| 78 | U. akitaensis | 8 | 2 | 0.-0.5 | 23,01 | 1,4 | 227,4 | 0,10 | |
| 78 | U. akitaensis | 8 | 6 | 0.-0.5 | 23,39 | 1,4 | 227,4 | 0,10 | |
| 79 | U. akitaensis | 8 | 4 | 0-0,5 | 22,02 | 1,4 | 227,4 | 0,10 | |
| 79 | U. akitaensis | 8 | 2 | 0.-0.5 | 24,39 | 1,4 | 227,4 | 0,11 | |
| 79 | U. akitaensis | 8 | 6 | 0.-0.5 | 46,83 | 1,4 | 227,4 | 0,21 | |
| 80 | U. akitaensis | 8 | 2 | 0.-0.5 | 6,28 | 1,4 | 227,4 | 0,03 | |
| 80 | U. akitaensis | 8 | 5 | 0.-0.5 | 12,29 | 1,4 | 227,4 | 0,05 | |
| 81 | U. akitaensis | 8 | 4 | 0-0,5 | 27,20 | 1,4 | 227,4 | 0,12 | |
| 81 | U. akitaensis | 8 | 2 | 0.-0.5 | 24,41 | 1,4 | 227,4 | 0,11 | |
| 82 | U. akitaensis | 8 | 2 | 0.-0.5 | 17,27 | 1,4 | 227,4 | 0,08 | |
| 82 | U. akitaensis | 8 | 6 | 0.-0.5 | 18,29 | 1,4 | 227,4 | 0,08 | |
| 83 | U. akitaensis | 8 | 4 | 0-0,5 | 26,71 | 1,4 | 227,4 | 0,12 | |
| 83 | U. akitaensis | 8 | 6 | 0.-0.5 | 18,49 | 1,4 | 227,4 | 0,08 | |
| 13 | U. akitaensis | 9 | 1 | 1-1,5 | 63,84 | 1 | no data | | |
| 13 | U. akitaensis | 9 | 3 | 1-1,5 | 41,44 | 1 | no data | | |
| 13 | U. akitaensis | 9 | 5 | 1-1,5 | 175,96 | 1 | no data | | |
| 14 | U. akitaensis | 9 | 1 | 1-1,5 | 29,24 | 1 | no data | | |
| 14 | U. akitaensis | 9 | 3 | 1-1,5 | 26,91 | 1 | no data | | |
| 14 | U. akitaensis | 9 | 6 | 1-1,5 | 79,43 | 1 | no data | | |
| 15 | U. akitaensis | 9 | 1 | 1-1,5 | 55,41 | 1 | no data | | |
| 15 | U. akitaensis | 9 | 3 | 1-1,5 | 60,78 | 1 | no data | | |
| 15 | U. akitaensis | 9 | 5 | 1-1,5 | 87,37 | 1 | no data | | |
| 84 | U. akitaensis | 9 | 4 | 0-0,5 | 29,65 | 1 | no data | | |
| 84 | U. akitaensis | 9 | 6 | 0.-0.5 | 45,43 | 1 | no data | | |
| 85 | U. akitaensis | 9 | 4 | 0-0,5 | 46,63 | 1 | no data | | |
| 85 | U. akitaensis | 9 | 2 | 0.-0.5 | 77,71 | 1 | no data | | |
| 85 | U. akitaensis | 9 | 6 | 0.-0.5 | 62,51 | 1 | no data | | |
| 86 | U. akitaensis | 9 | 4 | 0-0,5 | 24,45 | 1 | no data | | |
| 86 | U. akitaensis | 9 | 2 | 0.-0.5 | 61,07 | 1 | no data | | |
| 86 | U. akitaensis | 9 | 6 | 0.-0.5 | 67,69 | 1 | no data | | |
| 87 | U. akitaensis | 9 | 4 | 0-0,5 | 30,02 | 1 | no data | | |
| 87 | U. akitaensis | 9 | 2 | 0.-0.5 | 24,52 | 1 | no data | | |
| 87 | U. akitaensis | 9 | 6 | 0.-0.5 | 28,87 | 1 | no data | | |
| 88 | U. akitaensis | 9 | 4 | 0-0,5 | 104,08 | 1 | no data | | |
| 88 | U. akitaensis | 9 | 2 | 0.-0.5 | 20,10 | 1 | no data | | |
| 89 | U. akitaensis | 9 | 4 | 0-0,5 | 54,55 | 1 | no data | | |
| 89 | U. akitaensis | 9 | 2 | 0.-0.5 | 37,77 | 1 | no data | | |
| 89 | U. akitaensis | 9 | 6 | 0.-0.5 | 38,01 | 1 | no data | | |
| 90 | U. akitaensis | 9 | 2 | 0.-0.5 | 41,16 | 1 | no data | | |
| 90 | U. akitaensis | 9 | 6 | 0.-0.5 | 55,19 | 1 | no data | | |
| 91 | U. akitaensis | 9 | 2 | 0.-0.5 | 65,51 | 1 | no data | | |
| 91 | U. akitaensis | 9 | 6 | 0.-0.5 | 50,95 | 1 | no data | | |
| 92 | U. akitaensis | 9 | 2 | 0.-0.5 | 10,79 | 1 | no data | | |
| 92 | U. akitaensis | 9 | 6 | 0.-0.5 | 70,93 | 1 | no data | | |
| 93 | U. akitaensis | 9 | 2 | 0.-0.5 | 47,44 | 1 | no data | | |
| 93 | U. akitaensis | 9 | 6 | 0.-0.5 | 70,19 | 1 | no data | | |
| 94 | U. akitaensis | 9 | 5 | 0,5-1 | 91,65 | 1 | no data | | |
| 94 | U. akitaensis | 9 | 2 | 0.5-1 | 12,72 | 1 | no data | | |
| 94 | U. akitaensis | 9 | 6 | 0.5-1 | 54,61 | 1 | no data | | |
| 95 | U. akitaensis | 9 | 5 | 0,5-1 | 53,94 | 1 | no data | | |
| 95 | U. akitaensis | 9 | 2 | 0.5-1 | 172,32 | 1 | no data | | |
| 95 | U. akitaensis | 9 | 6 | 0.5-1 | 28,94 | 1 | no data | | |
| 96 | U. akitaensis | 9 | 4 | 0,5-1 | 64,88 | 1 | no data | | |
| 96 | U. akitaensis | 9 | 2 | 0.5-1 | 58,96 | 1 | no data | | |

| Specimen id | Species | Station | Chamber F= | Sed depth (cm) | Mn/Ca umol/mol | Average living depth cm | Mn/Ca$_{pw}$ at ALD umol/mol | D$_{Mn}$ | D$_{Mn}$ average per station |
|---|---|---|---|---|---|---|---|---|---|
| 11 | N. labradorica | 7 | 4 | 0-0,5 | 45,20 | 2,3 | no data | | |
| 9 | N. labradorica | 7 | 4 | 0-0,5 | 101,06 | 2,3 | no data | | |
| 42 | N. labradorica | 7 | 5 | 0-0,5 | 57,37 | 2,3 | no data | | |
| 43 | N. labradorica | 7 | 1 | 0-0,5 | 118,84 | 2,3 | no data | | |
| 43 | N. labradorica | 7 | 6 | 0-0,5 | 97,63 | 2,3 | no data | | |
| 14 | N. labradorica | 8 | 4 | 0-0,5 | 57,32 | 4 | 81,1 | 0,71 | 1,24 |
| 15 | N. labradorica | 8 | 1 | 0-0,5 | 50,31 | 4 | 81,1 | 0,62 | |
| 16 | N. labradorica | 8 | 3 | 2-2,5 | 23,40 | 4 | 81,1 | 0,29 | |
| 17 | N. labradorica | 8 | 4 | 2-2,5 | 63,18 | 4 | 81,1 | 0,78 | |
| 18 | N. labradorica | 8 | 1 | 4,0-5,0 | 276,98 | 4 | 81,1 | 3,41 | |
| 21 | N. labradorica | 8 | 4 | 4,0-5,0 | 157,87 | 4 | 81,1 | 1,95 | |
| 22 | N. labradorica | 8 | 3 | 4,0-5,0 | 189,49 | 4 | 81,1 | 2,34 | |
| 23 | N. labradorica | 8 | 4 | 4,0-5,0 | 35,09 | 4 | 81,1 | 0,43 | |
| 25 | N. labradorica | 8 | 4 | 4,0-5,0 | 54,58 | 4 | 81,1 | 0,67 | |
| 26 | N. labradorica | 10 | 3 | 0-0,5 | 27,58 | 4 | 391,3 | 0,070 | 0,20 |
| 27 | N. labradorica | 10 | 3 | 0-0,5 | 51,45 | 4 | 391,3 | 0,131 | |
| 28 | N. labradorica | 10 | 4 | 0-0,5 | 72,33 | 4 | 391,3 | 0,185 | |
| 29 | N. labradorica | 10 | 1 | 0-0,5 | 169,29 | 4 | 391,3 | 0,433 | |

| Specimen id | Species | Station | Chamber F= | Sed depth (cm) | Mn/Ca umol/mol | Average living depth cm | Mn/Ca$_{pw}$ at ALD umol/mol | DMn | DMn average per station |
|---|---|---|---|---|---|---|---|---|---|
| 17 | C. fimbriata | 7 | 1 | 0-0,5 | 70,03 | 2,7 | no data | | |
| 19 | C. fimbriata | 7 | 1 | 2,5-3 | 124,65 | 2,7 | no data | | |
| 23 | C. fimbriata | 8 | 1 | 4,0-5,0 | 41,42 | 5,1 | 38,9 | 1,07 | 1,77 |
| 23 | C. fimbriata | 8 | 1 | 4,0-5,0 | 78,80 | 5,1 | 38,9 | 2,03 | |
| 24 | C. fimbriata | 8 | 1 | 4,0-5,0 | 86,32 | 5,1 | 38,9 | 2,22 | |
| 29 | C. fimbriata | 9 | 1 | 1-1,5 | 94,98 | 5 | no data | | |
| 30 | C. fimbriata | 9 | 1 | 1-1,5 | 81,85 | 5 | no data | | |
| 31 | C. fimbriata | 9 | 1 | 3-3,5 | 199,08 | 5 | no data | | |
| 32 | C. fimbriata | 9 | 1 | 3-3,5 | 228,21 | 5 | no data | | |
| 33 | C. fimbriata | 9 | 1 | 3-3,5 | 69,26 | 5 | no data | | |
| 34 | C. fimbriata | 9 | 1 | 3-3,5 | 95,37 | 5 | no data | | |
| 35 | C. fimbriata | 9 | 1 | 3-3,5 | 184,99 | 5 | no data | | |
| 36 | C. fimbriata | 9 | 1 | 3-3,5 | 231,29 | 5 | no data | | |
| 37 | C. fimbriata | 9 | 1 | 3-3,5 | 110,20 | 5 | no data | | |
| 38 | C. fimbriata | 9 | 1 | 3-3,5 | 86,94 | 5 | no data | | |

| Specimen id | Species | Station | Chamber F= | Sed depth (cm) | Mn/Ca umol/mol | Average living depth cm | Mn/Ca$_{pw}$ at ALD umol/mol | DMn | DMn average per station |
|---|---|---|---|---|---|---|---|---|---|

**PORE WATER**

| Station | sedm depth cm | mid depth cm | Mn umol/l |
|---|---|---|---|
| 10 | bottom water | bottom water | 0,1 |
| | 0-0.5 | 0,25 | 1,1 |
| | 0.5-1 | 0,75 | 1,7 |
| | 1-1.5 | 1,25 | 1,5 |
| | 1.5-2 | 1,75 | 1,6 |
| | 2-3 | 2,5 | 3,6 |
| | 3-4 | 3,5 | 3,9 |
| | 4-5 | 4,5 | 4,5 |
| | 5-6 | 5,5 | 5,0 |
| | 6-7 | 6,5 | 4,6 |
| | 8-9 | 8,5 | 4,5 |
| | 9-10 | 9,5 | 3,8 |
| | 10-12 | 11 | 2,9 |
| | 12-14 | 13 | 1,7 |
| | 14-16 | 15 | 1,3 |
| | 16-18 | 17 | 1,2 |
| | 18-20 | 19 | 1,1 |
| | | | |
| 8 | bottom water | bottom water | 0,1 |
| | 0-0.5 | 0,25 | 1,1 |
| | 0.5-1 | 0,75 | 2,1 |
| | 1-1.5 | 1,25 | 2,3 |
| | 1.5-2 | 1,75 | 1,6 |
| | 2-3 | 2,5 | 1,2 |
| | 3-4 | 3,5 | 0,8 |
| | 4-5 | 4,5 | 0,5 |
| | 5-6 | 5,5 | 0,4 |
| | 6-7 | 6,5 | 0,4 |
| | 7-8 | 7,5 | 0,4 |
| | 8-9 | 8,5 | 0,6 |
| | 9-10 | 9,5 | 0,4 |
| | 10-12 | 11 | 0,4 |
| | 12-14 | 13 | 0,4 |
| | 14-16 | 15 | 0,4 |
| | 16-18 | 17 | |
| | 18-20 | 19 | 0,9 |
| | | | |
| 6 | bottom water | bottom water | 0,0 |
| | 0-0.5 | 0,25 | 1,1 |
| | 0.5-1 | 0,75 | 1,4 |
| | 1-1.5 | 1,25 | 1,2 |
| | 1.5-2 | 1,75 | 0,6 |
| | 2-3 | 2,5 | 0,5 |
| | 3-4 | 3,5 | 1,0 |
| | 4-5 | 4,5 | 0,3 |
| | 5-6 | 5,5 | 0,3 |
| | 6-7 | 6,5 | 0,3 |
| | 7-8 | 7,5 | 0,3 |
| | 8-9 | 8,5 | 0,4 |
| | 9-10 | 9,5 | 0,2 |
| | 10-12 | 11 | 0,3 |
| | 12-14 | 13 | 0,4 |
| | 14-16 | 15 | 0,6 |
| | 16-18 | 17 | 0,6 |
| | 18-20 | 19 | 1,1 |