# Peer review of "Benthic foraminiferal Mn/Ca ratios reflect microhabitat preferences"

_Biogeosciences, 2016_

## Referee Comment (RC1) · 14 Feb 2017

The manuscript entitled "Benthic foraminiferal Mn/Ca ratios reflect microhabitat preferences" by Karoliina Koho and colleagues presents foraminiferal Mn/Ca as a potential tool for paleoceanographic recontructions of the microhabitat, bottom water oxygenation and/or Mn redox chemistry. The research is original and provides novel, interesting data about Mn incorporation into foraminifera for the community. The methods used are state of the art and well suited to answer the research questions posed, however, more details need to be provided concerning the ICPMS measurements, especially since two different ICPMS setups in combination with different signal integration techniques were used, to ensure comparability of the data. The presented data is of appropriate quality, however, foraminiferal Mn/Ca ratios are only represented in Figures and $D_{Mn}$ values only mentioned in the text so that I strongly encourage the authors to provide this data in tables. In the case of $D_{Mn}$ also in a Figure similar to Fig. 5. In a few cases, I cannot confirm drawn conclusions from the data presented here, an urge the authors to revise those statements (indicated below). Furthermore, I would like the authors to encourage to sharpen the manuscript, that those parameters influencing foraminiferal Mn are more clear.

Overall, this is a well written manuscript of an interesting study and I would recommend publication after major revisions have been carried out. I wish the authors good luck with the revisions and remain available for further feedback and discussions.

Best wishes,

Nina Keul

**Comments by page and line number:**

**major comments:**

**page 5: concerning the methods used**

l.1: How long were the measurements on the different species? How long was one cycle of the ICPMS through all masses? Were there any short measurements due to e.g. thin chambers? Where they discarded? How much of the profiles were left out of the integration windows in glitter due to contamination? How many data points were left after this procedure on average? Was the contamination in high Al limited to the beginning and ends of the profiles?

l.6: which mass was measured for Mg? (it is not in the list in l. 22)? Were high Mg and Mn and high Al always restricted to the same spot? Could you maybe provide a couple of ablation profiles in the appendix to illustrate this?

l.10/11: was there no matrix matched in-house standard measured? e.g. GJR or JCP? If not, why not as matrix matched standards are common practice and have been used on the second setup?

I do not understand, why the measurements were calcibrated against NIST610 values from

Jochum et al. 2000 on one machine and against Jochum et al., 2011 on the other machine? Also, in the Jochum et al. 2000 paper cited here I cannot find reported concentrations on NIST610?

l. 13: which samples on which machine? Were some samples measured on both systems to ensure comparability? This is especially of importance with the apparently two different NIST610calibration values used? Could this data be provided in a supplementary table?

l. 23: "consisted of a blank"? I assume the first 20 seconds the laser was not switched on so this was the background and not a blank? Also, so here values were integrated manually and not using Glitter, why?

**page 6**

l. 29: please provide table with Mn/Ca measurements (and also for calculated DMn for stations 6,8,10)

**page 8**

l. 3: was only  the correlation with station bottom depth stat. significant or were also other parameters tested? It is mentioned in the abstract and Conclusion that Mn/Ca could be a sensitive recorder of redox conditions and or bottom water oxygenation, so a statistical test of this would be highly valuable.

l.27/l.28: However, some specimen occurred at different depths in the sediment at the same core location, how likely is it, that they also calcify at the same depth (ideally the ALD) for the species to be a good proxy (e.g. B. spissa at station 9)? Since in the upper few cm, vast changes wrt redox chemistry occur, potentially influencing foraminiferal Mn/Ca.

**page 9**

l.19-29: since the $D_{Mn}$ values are not listed in the paper (no table and no figure) I cannot assess this part, please provide data

l. 29/ 30: "implies that these taxa are actively growing in dysoxic sediments…" *B. spissa* in station 9 also has high Mn values, similar to *C. fimbriata*, please discuss (high Mn values do the not necessarily exclusively occur in deep infaunal species?)

Also, please discuss:
 As shown from the Mn porewater profiles (Fig. 5) and since most of the sediment is dysoxic after 1 cm depth (Fig. 2), the porewater concentrations in Mn are very different between station 6 (more or less constant Mn), Stn. 8 (Mn maximum at ca. 10 mm) and station 10 (Mn increases with depth) so in my interpretation of the data high Mn does not necessarily indicate only dysoxic environments, since this is the case in all the calcification environments and must be the signature of some other parameter?

**page 10:**

l. 11" deeper in the sediment where higher Mn conc. are present": I do see the increase in Mn with depth only at station 10, not the others, so this statement in my opinion cannot be drawn.

"a clear increase in foraminiferal Mn is observed as well": In this case, it would be very valuable to show a regression of foraminiferal Mn to porewater Mn to underline this statement.

**page 11:**

section (4.3). should be revised- at the moment, the paleographic implications from the measurements presented here (Mn/Ca in foraminifera and Mn in porewaters), should be the main focus in addition to comparison to literature values (this part is included). However, the present version discusses the relevance of the Troxchem model at length in addition the Mn redoxchemistry, however, only very little focus lies on the paleo implications of this study. Please move the discussion of the TROXCHEM model and the redox chemistrz into a different paragraph.

Furthermore, I am having a hard time to discern the key messages of the study wrt to what influences foraminiferal Mn/Ca. I agree with conclusion, that deeper fauna displays higher Mn/Ca, and that the deeper species must be calcifying under dysoxic conditions, but from the data presented I am having a hard time to see that "Mn incorporation" reflects
1) bottom water oxygenation (where is the data- regressions/ statistics and or figures? e.g. regression of foram Mn and BWO) representing this?
2) Mn redox chemistry (where is the data? regressions? statistics)
3) no ontogenetic influence (as argued above, it could be that interspecies variability masks this, since on most specimen, only 2,3 chambers are measured. However, I am positive that data can be easily presented in a revised version to be able to make this statement.

**minor comments:**

**page 1**

l.16: calcium carbonate tests

l.19: define BWO or spell out; what are differences exactly?

l.20: where is this entangling happening in manuscript?

l.24: At each station, Mn/Ca  (omit "the")

    also Mn/Ca is a ratio of concentrations, not a concentration

l.31: the forams are not the tools, but carry the proxy -> rephrase

l.32: has a high…

**page 2**

l.4: have been shown to reflect carbonate chemistry (omit "the")

l.18: are oxygenated and sediments are anoxic… add "and sediments are anoxic"

l.27: omit "the" before shallow

l.28: than not then

l.32: why 33 (random?) also omit "the" before foraminifera and change to foraminiferal

**page 3**

l.17: change to sth. like this as it is confusing otherwise: "At each site, three separate…"

l.19: company that produced CTD (seabird?), what is the error of the oxygen microsensor? Is it also called a "micro"sensor when it is attached to a CTD?

l.25: Whole sample centrifuged or subsample?

l.30: how much HCl was added? final conc.? What samples were used for storage? Were they acid cleaned?

**page 4**

l.1: I assume cps were measured and then converted to conc. via a calibration curve for those elements measured on the ICPMS? What wavelengths were measured in the OES? Which elements were measured on which machine? Which isotopes were measured on the ICPMS?

l.5 - 12: As I am unfamiliar with the methods and the custom built incubation chamber please provide a few more details to clarify:

I assume the subsample taken with the syringe was analyzed? Stabilization of what? temp. and oxygen? How were the fluctuations in oxygen conc. assessed? Were the stabilization times similar between cores (ca. 9hrs)? Were the oxygen profiles taken continuously or at certain depths?

l.13: Change title so it is more precise: e.g. "Foraminifera: sampling an elemental concentrations"

l.14 et al.

l.17: Plummer slides? Are they micropaleoslides?

l.30: So if the crater is 80µm I assume all foraminiferal chambers measured are bigger than that to make sure, that only one chamber is ablated per measurement?

**page 5**

l. 26: NFHS: has the homogeneity of this standard been published somewhere? Were JCP21 MACS3 and NFHS all used as the form of pressed powder tablets?

l.30: I assume seawater= porewater? where is DMn reported?

Knowing the good quality of data that usually is published from the Utrecht setup used, I assume that the methods have been written up by two different co-authors, I would strongly encourage the authors to rewrite section 2.5 so that the same details are given for both setups used.

**page 6**

l.6: what exactly is pore water chemistry? which parameters?

l.8: in-sediment depth? what depth is this?

l. 23: App. 1 is missing, I contacted the first author for App. 1, the excel file I received looks like there was mostly 2-3 chambers measured on each specimen, so that I doubt that this is enough to support that "there is no correlation between shell size and Mn/Ca" as it could be that interspecies variability masked potential ontogenetic trends in Mn/Ca, if only 2 or 3 chambers were measured on one specimen. I would encourage the authors to provide a figure in the appendix to demonstrate intra-species variability and also to calculate inter- versus intra-species variability for all species studied and provide data in a table.
Also I do not see statistical analyses in App. 1 (L. 23: "The statistical analyses were carried out on all data (App.1)").

**page 7**
l. 28: lowest average (?) Mn/Ca values

**page 8:**
l.16: "excluded from data": show also in exemplary profile (see comment above)
"Due to the nature of the specimens…" does this refer to the fact that living foraminifera most likely do not have diagenetic coatings or some other factor?

**page 9:**
please add references to figures and tables (also the "new" one with the Mn/Ca and $D_{Mn}$ values)

l.15/16: bimodal distribution - which species here shows a bimodal distribution?

l. 26: delete "are"

**page 10:**

l. 20: fluxes must still ==be== relatively

l.32 remove "study" at end of sentence

**page 11:**

l.6: fig 2 not fig1

last paragraph: good discussion of Mn redox chemistry and availability, but maybe move up in the manuscript, as it is in general relevant for the incorporation of Mn into foraminifera and not necessarily part of the "paleo implications only".

---

## Referee Comment (RC2) · Anonymous Referee #2 · 16 Feb 2017

The manuscript "Benthic foraminiferal Mn/Ca ratios reflect microhabitat preferences" by Koho et al. presents new data on the link between pore water Mn concentrations which are related to dissolved oxygen content, and benthic foraminiferal Mn/Ca. Mn/Ca is receiving a lot of attention recently as it may be a suitable proxy to reconstruct past dissolved oxygen concentrations in the water column/pore water. Using several different species and linking the data with pore water measurements has resulted in a very nice dataset, which partly provides evidence for existing ideas but also points out some issues that still exist. Especially the discussion on these possible issues could still use some more extensive consideration as described below in detail. But in general, the manuscript is well-written, easy and clear to follow, and definitely fitting within the scope of Biogeosciences. I recommend that this manuscript is suitable for publication after moderate revisions have been made.

My main issue is that I feel that the discussion on the part where pore water Mn/oxygen and Mn/Ca in the forams are not fitting, can be explored further. Currently, it is partly contradicting, i.e. living labradorica and fimbriata were found at 0-1 cm but are generally deeper-living species (unless maybe in conditions where the bottom water is already close to anoxic), so that would imply habitat migration. But then the lack of a trend in Mn/Ca in the chambers would indeed point to no migration. In station 8, both species have the highest Mn/Ca again and are deepest, but there is no Mn in the pore water. So under the anoxic conditions all the available Mn has either diffused upwards when reduction took place or it precipitated as MnCO3. How then can the forams have high Mn/Ca? For me this either means that they did migrate and picked up the Mn at a shallower depth; or that pore water oxygen and thus Mn are changing through the seasons, having higher pore water Mn when the forams calcified (assuming they were not calcifying at the moment of collection); or finally that the test Mn/Ca is biased by MnCO3 precipitation. You did write that contamination on in- and outside bits (high Al and or Mn) was discarded, but it would be interested to know if especially in these deep station 8 forams there was indeed a Mn-coating. Because if a coating forms, crystals may as easily form somewhere inside the test to bias the bulk Mn/Ca.

Even though that in general the relation between oxygen and Mn/Ca seems to follow the expected trends, the species-specific correlations are not very good or non-existing. What do you think could be the reason for that? How could the impact of habitat migration be determined? Seasonality may be resolved of course by extra sampling, which is always welcome. As a side note, I do like to point out that it would have been great to have had pore water profiles for stations 7 and 9 too.

Minor Comments: 2.1 add some of the main currents and water masses to figure 1. p.4, 16: part of the previously mentioned loop of possible explanations why not everything fits. Could it be that some of the deeper specimens in the anoxic sediment are stained despite being dead? They would still classify as recently-alive, but that may be enough to have them buried a couple of cms. p.5, 6: Mg? Mg/Ca data would of

course also be interesting to present. But to stick to redox elements, were any other redox elements like Fe or U analyzed? p.5, 26: Internal reproducibility is good, but how was the comparison between both lasers? p.6, 23: shell size; can a trend in different chambers automatically be related to shell size? I am not sure if this is a correct way of naming it. p.7, 29: rations, delete n p.10, 20: where the TROXCHEM, add the; still be, add be p.12, 2: foraminferal table 2: change comma's for decimals to points. Figure 1: add currents and watermasses Figure 5 caption: in indicated, change to is Figure 6 caption: this is exactly the same as the one for figure 5, which I assume should not be the case.

---

## Author Comment (AC1) · 7 Apr 2017

Below we have copied the reviewers' comments one at the time and indicate how we have addressed them or (in a few cases) argue why we respectfully disagree.

Our resubmission contains a typed manuscript, which is accompanied by eight figures, three tables and two appendixes.

REVIEW 1: NINA KEUL

The manuscript entitled "Benthic foraminiferal Mn/Ca ratios reflect microhabitat preferences" by Karoliina Koho and colleagues presents foraminiferal Mn/Ca as a potential tool for paleoceanographic reconstructions of the microhabitat, bottom water oxygenation and/or Mn redox chemistry. The research is original and provides novel, interesting

data about Mn incorporation into foraminifera for the community. The methods used are state of the art and well suited to answer the research questions posed, however, more details need to be provided concerning the ICPMS measurements, especially since two different ICPMS setups in combination with different signal integration techniques were used, to ensure comparability of the data. The presented data is of appropriate quality, however, foraminiferal Mn/Ca ratios are only represented in Figures and DMn values only mentioned in the text so that I strongly encourage the authors to provide this data in tables. In the case of DMn also in a Figure similar to Fig. 5. In a few cases, I cannot confirm drawn conclusions from the data presented here, an urge the authors to revise those statements (indicated below). Furthermore, I would like the authors to encourage to sharpen the manuscript, that those parameters influencing foraminiferal Mn are more clear. Overall, this is a well written manuscript of an interesting study and I would recommend publication after major revisions have been carried out. I wish the authors good luck with the revisions and remain available for further feedback and discussions. Best wishes, Nina Keul

RESPONSE: The authors thank Nina Keul for thorough review of the manuscript. Her comments have substantially improved the new version of our manuscript. This version now contains an appendix with the Mn/Ca measurements and also the calculated DMn values. In addition, we added a figure, in which the DMn-values are presented. More details are now provided concerning the LA-ICPMS analyses, and the discussion concerning the parameters influencing foraminiferal Mn has been sharpened.

Comments by page and line number: major comments: page 5: concerning the methods used 1.1: How long were the measurements on the different species? How long was one cycle of the ICPMS through all masses? Were there any short measurements due to e.g. thin chambers? Where they discarded? How much of the profiles were left out of the integration windows in glitter due to contamination? How many data points were left after this procedure on average? Was the contamination in high Al limited to the beginning and ends of the profiles? RESPONSE: Short profiles (typically <5s

in length) and profiles containing high Al content contents were excluded from further analysis. A representative profile ranges from 10-30 seconds, depending on species and chamber ablated (final chambers are commonly thinner and hence result in shorter ablation profiles). Some species e.g. N. labradorica and C. fimbriata had relatively short profiles, as their chamber walls are on average thinner. In case of E. batialis longer profiles were obtained due to thicker test calcite. In total, 277 single-chambered measurements were used for our study (Table 2). A new figure (Figure 2) is included, now showing examples of ablation profiles for Al/Ca, Mn/Ca and Mg/Ca. Al contamination was typically observed at the start of the profile (i.e. at the outside surface of the foraminiferal shell), although a very small Al peak was occasionally seen also inside. Often coinciding with these Al-peaks, Mg and Mn contamination peaks were observed inside and outside of the test. The text on the post-processing of the LA-CIP-MS data has been extended by incorporating this information (section 2.5) Regarding a cycle length of the ICPM through all masses, these differ per ICP-MS and masses studied. In NIOZ measurements the cycle length was 0.12 seconds. In Utrecht measurements the cycle length was 0.64 seconds. These are now added to the methods section.

1.6: which mass was measured for Mg? (it is not in the list in 1. 22)? Were high Mg and Mn and high Al always restricted to the same spot? Could you maybe provide a couple of ablation profiles in the appendix to illustrate this? RESPONSE: For magnesium, we measured 24Mg and 26Mg. Two typical ablation profiles have now been added to the methods section. Mg, Mn and Al were usually elevated at the outer surface of the test. With Al, Mg and Mn often peaking at the same time (i.e. depth in ablation profile). A smaller Al peak was also occasionally also observed without the other elements being elevated. The text has been modified and these details have now been included into the manuscript.

1.10/11: was there no matrix matched in-house standard measured? e.g. GJR or JCP? If not, why not as matrix matched standards are common practice and have been used on the second setup? I do not understand, why the measurements were calibrated

against NIST610 values from Jochum et al. 2000 on one machine and against Jochum et al., 2011 on the other machine? Also, in the Jochum et al. 2000 paper cited here I cannot find reported concentrations on NIST610? RESPONSE: The reference to Jochum et al. (2000) was a mistake: all references to certified NIST values were supposed to be Jochum et al., 2011. This mistake is now corrected in the text. As already stated in the original version of our manuscript, we also ablated pressed powders JCp-1, MACS-3 and an in-house foraminiferal 'standard', the NFHS (Mezger et al., 2016) to monitor drift and detect any potential offsets caused by switching between matrices and between materials with varying element concentrations.

l.13: which samples on which machine? Were some samples measured on both systems to ensure comparability? This is especially of importance with the apparently two different NIST610calibration values used? Could this data be provided in a supplementary table? RESPONSE: Same NIST610 calibrations (Jochum et al., 2011) were used in both set ups. The mistake regarding "Jochum et al. (2000 versus 2011)" is now corrected. Some specimens were measured on both machines, namely E. batialis and Uvigerina spp. As LA-ICPMS is a destructive technique it is not possible to measure the exact same spot on the foraminifera shell twice, making the analyses not true replica's. Due to within specimen variability in elemental composition, Mn/Ca ratios are expected to vary slightly within specimens. However, as can been seen from comparing the overall distribution and elemental composition of E. batialis (Fig 4 and 5), the elemental composition is relatively consistent regardless of the laser ablation ICP-MS system used. This confirms that results from the two platforms are inter-changeable, not only for the standards used but also the samples themselves. A similar result was also published in De Nooijer et al. (2014a) for which three different systems were used and shown to result in comparable foraminiferal El/Ca. This reference is now cited in section 2.5

l.23: "consisted of a blank"? I assume the first 20 seconds the laser was not switched on so this was the background and not a blank? Also, so here values were integrated

manually and not using Glitter, why? RESPONSE: 'Blank' is replaced by 'background'. The use of data reduction software does not mean that the integration windows are not manually selected. The Glitter package was not available on the NWR/iCap platform, but instead the iCap's Qtegra software (Thermo Scientific) was used for data reduction. Both softwares are described in section 2.5, dealing with the laser ablation.

page 6 l. 29: please provide table with Mn/Ca measurements (and also for calculated DMn for stations 6,8,10) RESPONSE: Raw data now provided in appendix, which now also contains the calculated DMn-values.

page 8 l. 3: was only the correlation with station bottom depth stat. significant or were also other parameters tested? It is mentioned in the abstract and Conclusion that Mn/Ca could be a sensitive recorder of redox conditions and or bottom water oxygenation, so a statistical test of this would be highly valuable. RESPONSE: The correlation between Mn/Ca and station depth, as well as bottom water oxygenation (BWO) was tested for all taxa. The Mn/Ca ratios of Uvigerina spp. and B. spissa increased with water depth and were statistically significant (Section 3.4). Another statistically significant trend was found for Mn/Ca in B. spissa and BWO (Statistics are listed in Section 3.4). The correlations were tested for all taxa but these were the only significant trends within our dataset. Due to relatively few data points, correlations for deep infauna were very limited. In the results presented in Koho et al. (2015), a significant trend between Mn/Ca and BWO was also found for the intermediate infaunal species, Melonis barleeanus. Correlation of Mn/Ca in B. spissa and BWO imply that intermediate infaunal species may be the most prominent recorders of BWO in the setting studied here. This is consistent with the conclusion in Koho et al. (2015) based on the TROXCHEM3 model. Furthermore, although no statistical trends were found for other species and BWO, this study clearly illustrates systematic variations in Mn/Ca with foraminiferal microhabitat, coinciding with changes in pore water redox chemistry. Foraminifera inhabiting more oxygenated sediment layers (i.e. E. batialis) systematically showed low Mn/Ca ratios and more infaunal foraminiferal species, experiencing more reducing conditions, showed higher Mn/Ca ratios. Abstract and conclusions have now been adapted to reflect these outcomes.

l.27/l.28: However, some specimen occurred at different depths in the sediment at the same core location, how likely is it, that they also calcify at the same depth (ideally the ALD) for the species to be a good proxy (e.g. B. spissa at station 9)? Since in the upper few cm, vast changes wrt redox chemistry occur, potentially influencing foraminiferal Mn/Ca. RESPONSE: Yes it is true that foraminifera are not found only at their ALD and some specimens are found above and below this depth. The low DMn-values also suggests that for example E. batialis may calcify at slightly shallower depth than their inferred ALD suggests, whereas DMn- values of deep infaunal species (e.g. C. fimbricata) imply that it was found close to its calcification depth (Discussed in section 4.2). However, no systematic ontogenetic trends were seen in this study, where for example Mn/Ca ratios were systematically either higher or lower in the younger or older chambers, or vice versa. We observed no correlation between chamber number and Mn/Ca (See also new appendix 1). Still, variations in the overall Mn/Ca ratios for example in B. spissa shells may be due to migration or changes in the ambient pore water conditions during the lifespan of the foraminifera. The influence of migration on foraminiferal Mn/Ca is now discussed in section 4.2, 4th paragraph.

page 9 l.19-29: since the DMn values are not listed in the paper (no table and no figure) I cannot assess this part, please provide data RESPONSE: DMn values are now given in appendix 1

l. 29/ 30: "implies that these taxa are actively growing in dysoxic sediments..." B. spissa in station 9 also has high Mn values, similar to C. fimbriata, please discuss (high Mn values do the not necessarily exclusively occur in deep infaunal species?) RESPONSE: Unfortunately it is not possible to calculate DMn value for B. spissa at station 9 as no pore water Mn data is available for this station. At station 8 the DMn was low (0.36), suggesting that B. spissa is calcifying shallower than were it was found here. However, it is true that at station 9 the Mn/Ca ratios of B. spissa are similar to

that of deep infaunal species. This is now added to discussion.

Also, please discuss: As shown from the Mn porewater profiles (Fig. 5) and since most of the sediment is dysoxic after 1 cm depth (Fig. 2), the porewater concentrations in Mn are very different between station 6 (more or less constant Mn), Stn. 8 (Mn maximum at ca. 10 mm) and station 10 (Mn increases with depth) so in my interpretation of the data high Mn does not necessarily indicate only dysoxic environments, since this is the case in all the calcification environments and must be the signature of some other parameter? RESPONSE: It is true that pore water Mn concentrations are variable between stations and that the highest concentrations are found at the station 10 where bottom waters are relatively well ventilated. We suggest that this is due to variations in the availability of the Mn-oxides between the stations. This (and its implications for paleostudies) is discussed in depth in section 4.3 (3rd paragraph). At the deepest station (with the highest BWO content) Mn-oxides are accumulating in the sediments. As BWO content is relatively high, Mn is trapped and not able to escape into the water column, which may possibly occur at station 6 and 8. In addition it is possible that Mn-oxides are transported along slope, hence accumulating at the deeper and more ventilated areas. Although at each station some Mn-reduction was taking place, as shown by subsurface peaks in Mn, the pool of total dissolved Mn is likely to be related to availability of Mn-oxides in the sediment.

page 10: l. 11" deeper in the sediment where higher Mn conc. are present": I do see the increase in Mn with depth only at station 10, not the others, so this statement in my opinion cannot be drawn. "a clear increase in foraminiferal Mn is observed as well": In this case, it would be very valuable to show a regression of foraminiferal Mn to porewater Mn to underline this statement. RESPONSE: Indeed the largest increase in the pore water Mn with sediment depth is observed at station 10. At the other stations Mn-reduction is more limited, however, a small dissolved Mn- peak is also present at station 6 (at depth of 0.8 cm) and 8 (at depth of 1.3 cm), implying that Mn-reduction occurs in the sediment. Nevertheless, at station 10, where a clear peak in

Mn is present this is also reflected in the foraminiferal Mn/Ca ratios, as we see very low concentrations in the surface dwelling E. batialis and high concentrations in deep living N. labradorica. The sentence is now modified to make clear that this refers specifically to station 10. In addition, "clear" is omitted. Unfortunately species-specific regressions of foraminiferal Mn to pore water Mn are not possible due to limited pore water data.

page 11: section (4.3). should be revised- at the moment, the paleographic implications from the measurements presented here (Mn/Ca in foraminifera and Mn in porewaters), should be the main focus in addition to comparison to literature values (this part is included). However, the present version discusses the relevance of the Troxchem model at length in addition the Mn redox chemistry, however, only very little focus lies on the paleo implications of this study. Please move the discussion of the TROXCHEM model and the redox chemistry into a different paragraph. RESPONSE: Section 4.3 has been revised and the main paragraph dealing with the TROXCHEM model has been moved to section 4.2. However, we have decided to keep the discussion on Mn-redox chemistry as part of 4.3 as it has direct implications for the application of Mn/Ca down core. Our data shows that foraminiferal Mn/Ca is not only reflecting redox conditions but is also influenced by availability of Mn-oxides and hence Mn to be potentially released upon reduction. The availability of Mn-oxides, and hence the MnOx-reduction potential of the sediment, as shown by our results, is recorded in the foraminifera along our study transects. Therefore, paleoceanographic studies should take into account changes in the supply of Mn-oxides as well as changes in sediment oxygenation, as the former is also an important parameter in regulating pore water Mn-concentrations.

Furthermore, I am having a hard time to discern the key messages of the study wrt to what influences foraminiferal Mn/Ca. I agree with conclusion, that deeper fauna displays higher Mn/Ca, and that the deeper species must be calcifying under dysoxic conditions, but from the data presented I am having a hard time to see that "Mn incorporation" reflects RESPONSE: conclusions have been modified 1)bottom water oxygenation (where is the data- regressions/ statistics and or figures? e.g. regression of

foram Mn and BWO) representing this? RESPONSE: A statistically significant correlation was observed between BWO and Mn/Ca ratios in B. spissa (Results: section 3.4). Also statistically significant increases in the Mn/Ca ratios were observed along the study transect for Uvigerina spp. and B. spissa (Results: section 3.4). No regression analyses were carried as there is not sufficient pore water data (i.e. each species was not present at all three sites with available pore water data) to support this.

2) Mn redox chemistry (where is the data? regressions? statistics) RESPONSE: Mn pore water data was collected at three stations only and maximum of 2 taxa were measured at stations with pore water Mn-data. Hence unfortunately there is not sufficient Mn pore water data to carry out such analyses. Conclusions have now been modified to reflect this.

3) no ontogenetic influence (as argued above, it could be that interspecies variability masks this, since on most specimen, only 2,3 chambers are measured. However, I am positive that data can be easily presented in a revised version to be able to make this statement. RESPONSE: the reviewer likely refers to intraspecies variability, not interspecies variability. It is true that most specimens of Uvigerina spp. were measured two or three times, however, this is not true for B. spissa, which was measured 4 times in 14 out of 23 specimens. In addition, E. batialis was measured twice 5 times and once 6 times. None of these specimens showed systematic, statistically significant ontogenetic trends (now added to the appendix 1). Therefore, we can further conclude that no ontogentic trends were observed in our data. This is now clarified in results section 3.2. with a reference to Appendix 1.

minor comments: page 1 l.16: calcium carbonate tests RESPONSE: ok, done l.19: define BWO or spell out; what are differences exactly? RESPONSE: ok, done l.20: where is this entangling happening in manuscript? RESPONSE: changed "further resolving" l.24: At each station, Mn/Ca (omit "the") RESPONSE: ok, done also Mn/Ca is a ratio of concentrations, not a concentration RESPONSE: ok, changed to ratio l.31: the forams are not the tools, but carry the proxy -> rephrase RESPONSE: changed to

proxies used in paleoceanographic studies. l.32: has a high... RESPONSE: ok, done

page 2 l.4: have been shown to reflect carbonate chemistry (omit "the") RESPONSE: ok, done l.18: are oxygenated and sediments are anoxic... add "and sediments are anoxic" RESPONSE: changed to "In sediments, where bottom waters and surficial sediments are oxygenated and deeper sediments are anoxic..." l.27: omit "the" before shallow RESPONSE: ok, done l.28: than not then RESPONSE: ok, done l.32: why 33 (random?) also omit "the" before foraminifera and change to foraminiferal RESPONSE: 33 is not random this is the lowest BWO content along the study transect, so BWO was always higher than this. "The" is now omitted.

page 3 l.17: change to sth. like this as it is confusing otherwise: "At each site, three separate..." RESPONSE: "Separate cores were collected for pore water- and foraminiferal analyses, and oxygen profiling, all of which were derived from the same multicore cast." l.19: company that produced CTD (seabird?), what is the error of the oxygen microsensor? Is it also called a "micro"sensor when it is attached to a CTD? RESPONSE: The CTD is SBE9plus (Sea-Bird Electronics, S/N 860) and it was equipped with SBE3 thermometer (S/N 4378), SBE4 conductivity sensor (S/N 3307) and SBE43 oxygen sensor (S/N 0781). The details of the equipment are now added into the manuscript (section 2.2). "Oxygen microsensor" replaced with "oxygen sensor". The accuracy specifications of the oxygen sensor are typically within 2% of true value

l.25: Whole sample centrifuged or subsample? RESPONSE: Changed to "Sediment samples were centrifuged..." In general the whole samples were centrifuged. In case of the deepest sediment intervals where the slice thickness was 2 cm, some sediment may have been disregarded. l.30: how much HCl was added? final conc.? What samples were used for storage? Were they acid cleaned? RESPONSE: Text modified "Samples for pore water elemental analyses were acidified with suprapur HCl 37% ($10\mu l$ per ml of sample) and subsequently stored at 4°C until analyses at Utrecht University." The foraminiferal samples were not cleaned other than was indicated in the

original manuscript.

page 4 l.1: I assume cps were measured and then converted to conc. via a calibration curve for those elements measured on the ICPMS? What wavelengths were measured in the OES? Which elements were measured on which machine? Which isotopes were measured on the ICPMS? RESPONSE: Sentence modified to "Seawater elemental concentrations of 55Mn were measured with an inductively coupled plasma-mass spectrometer (ICP-MS, ThermoFisher Scientific Element2-XR)." Part about the OES is deleted as only ICP-MS data is reported in this article.

l.5 - 12: As I am unfamiliar with the methods and the custom built incubation chamber please provide a few more details to clarify: I assume the subsample taken with the syringe was analyzed? Stabilization of what? temp. and oxygen? How were the fluctuations in oxygen conc. assessed? Were the stabilization times similar between cores (ca. 9hrs)? Were the oxygen profiles taken continuously or at certain depths? RESPONSE: The O2 profiles have been published previously in Fontanier et al. (2014) with details of the employed methods. A citation has been added here. Each time a core was left to stabilize under insitu O2 and temperature conditions for 9hrs (as these parameters are likely to change during core recovery). O2 conditions were monitored with a microsensor, with no syringe being used. O2 profiles were made at 100$\mu$m resolution.

l.13: Change title so it is more precise: e.g. "Foraminifera: sampling an elemental concentrations" RESPONSE: changed to "Foraminifera: sampling and elemental composition"

l.14 et al. RESPONSE: ok done

l.17: Plummer slides? Are they micropaleoslides? RESPONSE: ok done

l.30: So if the crater is 80$\mu$m I assume all foraminiferal chambers measured are bigger than that to make sure, that only one chamber is ablated per measurement? RE-

SPONSE: The word "generally" was added. In general always single chamber was ablated, however, it can not be completely excluded that in few rare cases two chambers were ablated at once. This would mainly concern small individuals of B. spissa, which has very small older chambers.

page 5 l. 26: NFHS: has the homogeneity of this standard been published somewhere? Were JCP21 MACS3 and NFHS all used as the form of pressed powder tablets? RESPONSE: Yes: all these CaCO3 powders were pressed into tablets. The relative standard deviation in element/Ca based on multiple measurements on the NFHS is comparable to that of other standards (Mezger et al., 2016). A reference to this study is added (section 2.5 5th paragraph)

l.30: I assume seawater= porewater? where is DMn reported? Knowing the good quality of data that usually is published from the Utrecht setup used, I assume that the methods have been written up by two different co-authors, I would strongly encourage the authors to rewrite section 2.5 so that the same details are given for both setups used. RESPONSE: DMn values are all given now in Appendix 2. Yes, seawater changed to pore water. We have carefully checked section 2.5 to check for consistency and modifications have been made. However, the differences in setups and controls of the ICP-MS's used inherently cause the descriptions to differ. For example, tuning of the quadrupole and SF-ICP-MS differ and also the way they cycle through the elements analysed differs fundamentally. Still, all relevant parameters are described in each of the paragraphs.

page 6 l.6: what exactly is pore water chemistry? which parameters? RESPONSE: Sentence modified to "Pore water chemistry, including dissolved oxygen, nitrate, ammonium and manganese, was measured at sites 6, 8 and 10 (Figure 2)."

l.8: in-sediment depth? what depth is this? RESPONSE: Sentence modified to" In all cores, nitrate was rapidly depleted within surficial sediments"

l. 23: App. 1 is missing, I contacted the first author for App. 1, the excel file I received looks like there was mostly 2-3 chambers measured on each specimen, so that I doubt that this is enough to support that "there is no correlation between shell size and Mn/Ca" as it could be that interspecies variability masked potential ontogenetic trends in Mn/Ca, if only 2 or 3 chambers were measured on one specimen. I would encourage the authors to provide a figure in the appendix to demonstrate intra-species variability and also to calculate inter- versus intra-species variability for all species studied and provide data in a table. Also I do not see statistical analyses in App. 1 (L. 23: "The statistical analyses were carried out on all data (App.1)"). RESPONSE Appendix is now added, containing all raw data. This is labeled now as appendix 2. In addition, appendix 1 is now supplied where profiles of B. spissa and E. batialis are shown, based on Mn/Ca ratios in single specimens. Pearson correlation coefficients were calculated, however, all correlations were insignificant with two-tailed significance always being >0.05. This is added to results section 3.2.. Please also consider the response on this same topic earlier in the review.

page 7 l. 28: lowest average (?) Mn/Ca values RESPONSE: Nina Keul was contacted regarding this comment, as it was not clear to the authors what she originally meant with it. In her response she stated " I was wondering whether average shell Mn/Ca is in station 6 the lowest for that species or whether it is actually lower in shells from the same species in station 8, where porewater Mn was higher? (Sorry for the kryptic comment...) If that's the case it should be discussed somewhere. Our response to this comment: The average Mn/Ca ratio at station 6 is 30,1 and at station 8 30,6, so it is little bit higher at station 8 than 6, as should be due to slightly higher pore water Mn-content at station 8. However, the difference is not statistically significant. Based on this comment, it was noted that the standard error is missing in Figure 8 for Uvigerina spp (station 8) this is now added to the figure. For station 7 the error bar is so small (1,7) that it is hidden under the data label, and thus hardly visible.

page 8 l.16: "excluded from data": show also in exemplary profile (see comment above) "Due to the nature of the specimens. . ." does this refer to the fact that living foraminifera

most likely do not have diagenetic coatings or some other factor? RESPONSE: representative laser ablation profiles are now provided (figure 2). Indeed as specimens were very recent, diagenetic coatings are unlikely. This is now added into the sentence.

page 9: please add references to figures and tables (also the "new" one with the Mn/Ca and DMn values) RESPONSE: new figure with DMn made and referred to. Values also given in Appendix 2.

l.15/16: bimodal distribution - which species here shows a bimodal distribution? RESPONSE: B. spissa at station 7 and Uvigerina spp. station 7 and 8 (Fontanier et al. 2014). This is now added into the sentence.

l. 26: delete "are" RESPONSE: Modified to ". . .it seems that deep infaunal foraminifera, based on their Mn incorporation, are calcifying. . ."

page 10: l. 20: fluxes must still be relatively RESPONSE: ok, done

l.32 remove "study" at end of sentence RESPONSE: ok, done. This section has also now moved up into the start of section 4.3

page 11: l.6: fig 2 not fig1 RESPONSE: reference to figure corrected.

Last paragraph: good discussion of Mn redox chemistry and availability, but maybe move upin the manuscript, as it is in general relevant for the incorporation of Mn into foraminifera and not necessarily part of the "paleo implications only". RESPONSE: As outlined earlier in the response to the review, we feel that this has direct implications for paleo studies as it shows that the Mn/Ca ratios in foraminfera do not only depend on oxygenation, or redox chemistry, but also on supply of manganese oxides in sediment. Therefore, we have kept this section as part of 4.3.

—END OF REVIEW 1——

Please also note the supplement to this comment:
http://www.biogeosciences-discuss.net/bg-2016-547/bg-2016-547-AC1-

supplement.pdf

[Figure]

**Fig. 1.** A Regional map of the study area B: Bathymetric map of the study region, showing the position of Tsugaru warm current (Oguma et al., 2002) and multicore sampling sites. C: Schematized study

**Fig. 2.** Laser ablation profile for Al/Ca Mg/Ca and Mn/Ca measured in (A) E. baliatils (station 8, 0-0.5 cm depth) and (B) Uvigerina akitaensis (station 7, 0-0.5 cm depth) benthic foraminifera The se

**Fig. 3.** Pore water profiles of dissolved oxygen, nitrate, ammonium and manganese at station 6 (A), 8 (B) and 10 (C). (D) Pore water manganese inventory in the top 10 cm of sediment and bottom water oxygen con

[Figure]

**Fig. 4.** Figure 4: Box-plots showing chamber-to-chamber variability of Mn/Ca. Error bars display the full range of data variation (from minimum to maximum). Data outliers are represented with an astrix.

[Figure]

**Fig. 5.** Figure 5: Individual laser ablation measurements of Mn/Ca in foraminifera versus sediment depth where the specimens were collected.

[Figure]

**Fig. 6.** Mn/Ca ratios in foraminifera as a function of the average living depth of each species. The average of all measurements is indicated with a solid symbol and the individual measurements with open sym

[Figure]

**Fig. 7.** Manganese partition coefficient DMn in foraminifera as a function of average living depth of each species. In addition, the pore water (pw) Mn/Ca profile is shown.

[Figure]

**Fig. 8.** Variability of average Mn/Ca ratios of each species plotted against the study transect from station 6 to station 10 (left), and along the bottom water oxygenation (right). The error bars represen...

**Supplement:**

Dr. K. A. Koho
University of Helsinki
Department of Environmental Sciences
Viikinkaari 1
00790 Helsinki
Finland
Email: karoliina.koho@helsinki.fi

Helsinki, 7 April 2017

Dear Editor,

Hereby we resubmit our manuscript titled "Benthic foraminiferal Mn/Ca ratios reflect microhabitat preferences" for publication in Biogeosciences. An earlier version of this manuscript was reviewed by two referees all stating it is suitable for publication in the journal, providing that major or moderate changes are made to improve the paper. On the following pages we have copied the reviewers' comments one at the time and indicate how we addressed them or (in a few cases) argue why we respectfully disagree.

Our resubmission contains a typed manuscript, which is accompanied by eight figures, three tables and two appendixes.

Thank you for considering our submission for publication in Biogeosciences.

Sincerely,

Karoliina Koho, also on behalf of all co-authors

**REVIEW 1: NINA KEUL**

The manuscript entitled "Benthic foraminiferal Mn/Ca ratios reflect microhabitat preferences" by Karoliina Koho and colleagues presents foraminiferal Mn/Ca as a potential tool for paleoceanographic reconstructions of the microhabitat, bottom water oxygenation and/or Mn redox chemistry. The research is original and provides novel, interesting data about Mn incorporation into foraminifera for the community. The methods used are state of the art and well suited to answer the research questions posed, however, more details need to be provided concerning the ICPMS measurements, especially since two different ICPMS setups in combination with different signal integration techniques were used, to ensure comparability of the data. The presented data is of appropriate quality, however, foraminiferal Mn/Ca ratios are only represented in Figures and DMn values only mentioned in the text so that I strongly encourage the authors to provide this data in tables. In the case of DMn also in a Figure similar to Fig. 5. In a few cases, I cannot confirm drawn conclusions from the data presented here, an urge the authors to revise those statements (indicated below). Furthermore, I would like the authors to encourage to sharpen the manuscript, that those parameters influencing foraminiferal Mn are more clear. Overall, this is a well written manuscript of an interesting study and I would recommend publication after major revisions have been carried out. I wish the authors good luck with the revisions and remain available for further feedback and discussions.
Best wishes,
Nina Keul

*RESPONSE: The authors thank Nina Keul for thorough review of the manuscript. Her comments have substantially improved the new version of our manuscript. This version now contains an appendix with the Mn/Ca measurements and also the calculated $D_{Mn}$ values. In addition, we added a figure, in which the $D_{Mn}$-values are presented. More details are now provided concerning the LA-ICPMS analyses, and the discussion concerning the parameters influencing foraminiferal Mn has been sharpened.*

**Comments by page and line number:**
**major comments:**
page 5: concerning the methods used
1.1: How long were the measurements on the different species? How long was one cycle of the ICPMS through all masses? Were there any short measurements due to e.g. thin chambers? Where they discarded? How much of the profiles were left out of the integration windows in glitter due to contamination? How many data points were left after this procedure on average? Was the contamination in high Al limited to the beginning and ends of the profiles?
*RESPONSE: Short profiles (typically <5s in length) and profiles containing high Al content contents were excluded from further analysis. A representative profile ranges from 10-30 seconds, depending on species and chamber ablated (final chambers are commonly thinner and hence result in shorter ablation profiles). Some species e.g. N. labradorica and C. fimbriata had relatively short profiles, as their chamber walls are on average thinner. In case of E. batialis longer profiles were obtained due to thicker test calcite.*
*In total, 277 single-chambered measurements were used for our study (Table 2). A new figure (Figure 2) is included, now showing examples of ablation profiles for Al/Ca, Mn/Ca and Mg/Ca. Al contamination was typically observed at the start of the profile (i.e. at the outside surface of the foraminiferal shell), although a very small Al peak was occasionally seen also inside. Often coinciding with these Al-peaks, Mg and Mn contamination peaks were observed inside and outside of the test. The text on the post-processing of the LA-CIP-MS data has been extended by incorporating this information (section 2.5)*

*Regarding a cycle length of the ICPM through all masses, these differ per ICP-MS and masses studied. In NIOZ measurements the cycle length was 0.12 seconds. In Utrecht measurements the cycle length was 0.64 seconds. These are now added to the methods section.*

1.6: which mass was measured for Mg? (it is not in the list in 1. 22)? Were high Mg and Mn and high Al always restricted to the same spot? Could you maybe provide a couple of ablation profiles in the appendix to illustrate this?
*RESPONSE: For magnesium, we measured $^{24}$Mg and $^{26}$Mg. Two typical ablation profiles have now been added to the methods section. Mg, Mn and Al were usually elevated at the outer surface of the test. With Al, Mg and Mn often peaking at the same time (i.e. depth in ablation profile). A smaller Al peak was also occasionally also observed without the other elements being elevated. The text has been modified and these details have now been included into the manuscript.*

1.10/11: was there no matrix matched in-house standard measured? e.g. GJR or JCP? If not, why not as matrix matched standards are common practice and have been used on the second setup? I do not understand, why the measurements were calibrated against NIST610 values from Jochum et al. 2000 on one machine and against Jochum et al., 2011 on the other machine? Also, in the Jochum et al. 2000 paper cited here I cannot find reported concentrations on NIST610?
*RESPONSE: The reference to Jochum et al. (2000) was a mistake: all references to certified NIST values were supposed to be Jochum et al., 2011. This mistake is now corrected in the text. As already stated in the original version of our manuscript, we also ablated pressed powders JCp-1, MACS-3 and an in-house foraminiferal 'standard', the NFHS (Mezger et al., 2016) to monitor drift and detect any potential offsets caused by switching between matrices and between materials with varying element concentrations.*

l.13: which samples on which machine? Were some samples measured on both systems to ensure comparability? This is especially of importance with the apparently two different NIST610calibration values used? Could this data be provided in a supplementary table?
*RESPONSE: Same NIST610 calibrations (Jochum et al., 2011) were used in both set ups. The mistake regarding "Jochum et al. (2000 versus 2011)" is now corrected. Some specimens were measured on both machines, namely E. batialis and Uvigerina spp. As LA-ICPMS is a destructive technique it is not possible to measure the exact same spot on the foraminifera shell twice, making the analyses not true replica's. Due to within specimen variability in elemental composition, Mn/Ca ratios are expected to vary slightly within specimens. However, as can been seen from comparing the overall distribution and elemental composition of E. batialis (Fig 4 and 5), the elemental composition is relatively consistent regardless of the laser ablation ICP-MS system used. This confirms that results from the two platforms are inter-changeable, not only for the standards used but also the samples themselves. A similar result was also published in De Nooijer et al. (2014a) for which three different systems were used and shown to result in comparable foraminiferal El/Ca. This reference is now cited in section 2.5*

l.23: "consisted of a blank"? I assume the first 20 seconds the laser was not switched on so this was the background and not a blank? Also, so here values were integrated manually and not using Glitter, why?
*RESPONSE: 'Blank' is replaced by 'background'. The use of data reduction software does not mean that the integration windows are not manually selected. The Glitter package was not available on the NWR/iCap platform, but instead the iCap's Qtegra software (Thermo Scientific) was used for data reduction. Both softwares are described in section 2.5, dealing with the laser ablation.*

page 6
l. 29: please provide table with Mn/Ca measurements (and also for calculated DMn for stations 6,8,10)
*RESPONSE: Raw data now provided in appendix, which now also contains the calculated $D_{Mn}$-values.*

page 8
l. 3: was only the correlation with station bottom depth stat. significant or were also other parameters tested? It is mentioned in the abstract and Conclusion that Mn/Ca could be a sensitive recorder of redox conditions and or bottom water oxygenation, so a statistical test of this would be highly valuable.
*RESPONSE: The correlation between Mn/Ca and station depth, as well as bottom water oxygenation (BWO) was tested for all taxa. The Mn/Ca ratios of Uvigerina spp. and B. spissa increased with water depth and were statistically significant (Section 3.4). Another statistically significant trend was found for Mn/Ca in B. spissa and BWO (Statistics are listed in Section 3.4). The correlations were tested for all taxa but these were the only significant trends within our dataset. Due to relatively few data points, correlations for deep infauna were very limited. In the results presented in Koho et al. (2015), a significant trend between Mn/Ca and BWO was also found for the intermediate infaunal species, Melonis barleeanus.*
*Correlation of Mn/Ca in B. spissa and BWO imply that intermediate infaunal species may be the most prominent recorders of BWO in the setting studied here. This is consistent with the conclusion in Koho et al. (2015) based on the TROXCHEM[3] model. Furthermore, although no statistical trends were found for other species and BWO, this study clearly illustrates systematic variations in Mn/Ca with foraminiferal microhabitat, coinciding with changes in pore water redox chemistry. Foraminifera inhabiting more oxygenated sediment layers (i.e. E. batialis) systematically showed low Mn/Ca ratios and more infaunal foraminiferal species, experiencing more reducing conditions, showed higher Mn/Ca ratios. Abstract and conclusions have now been adapted to reflect these outcomes.*

l.27/l.28: However, some specimen occurred at different depths in the sediment at the same core location, how likely is it, that they also calcify at the same depth (ideally the ALD) for the species to be a good proxy (e.g. B. spissa at station 9)? Since in the upper few cm, vast changes wrt redox chemistry occur, potentially influencing foraminiferal Mn/Ca.
*RESPONSE: Yes it is true that foraminifera are not found only at their ALD and some specimens are found above and below this depth. The low $D_{Mn}$-values also suggests that for example E. batialis may calcify at slightly shallower depth than their inferred ALD suggests, whereas $D_{Mn}$- values of deep infaunal species (e.g. C. fimbricata) imply that it was found close to its calcification depth (Discussed in section 4.2). However, no systematic ontogenetic trends were seen in this study, where for example Mn/Ca ratios were systematically either higher or lower in the younger or older chambers, or vice versa. We observed no correlation between chamber number and Mn/Ca (See also new appendix 1). Still, variations in the overall Mn/Ca ratios for example in B. spissa shells may be due to migration or changes in the ambient pore water conditions during the lifespan of the foraminifera. The influence of migration on foraminiferal Mn/Ca is now discussed in section 4.2, 4th paragraph.*

page 9
l.19-29: since the DMn values are not listed in the paper (no table and no figure) I cannot assess this part, please provide data
*RESPONSE: $D_{Mn}$ values are now given in appendix 1*

l. 29/ 30: "implies that these taxa are actively growing in dysoxic sediments…" B. spissa in station 9 also has high Mn values, similar to C. fimbriata, please discuss (high Mn values do the not necessarily exclusively occur in deep infaunal species?)

*RESPONSE: Unfortunately it is not possible to calculate $D_{Mn}$ value for B. spissa at station 9 as no pore water Mn data is available for this station. At station 8 the $D_{Mn}$ was low (0.36), suggesting that B. spissa is calcifying shallower than were it was found here. However, it is true that at station 9 the Mn/Ca ratios of B. spissa are similar to that of deep infaunal species. This is now added to discussion.*

Also, please discuss: As shown from the Mn porewater profiles (Fig. 5) and since most of the sediment is dysoxic after 1 cm depth (Fig. 2), the porewater concentrations in Mn are very different between station 6 (more or less constant Mn), Stn. 8 (Mn maximum at ca. 10 mm) and station 10 (Mn increases with depth) so in my interpretation of the data high Mn does not necessarily indicate only dysoxic environments, since this is the case in all the calcification environments and must be the signature of some other parameter?

*RESPONSE: It is true that pore water Mn concentrations are variable between stations and that the highest concentrations are found at the station 10 where bottom waters are relatively well ventilated. We suggest that this is due to variations in the availability of the Mn-oxides between the stations. This (and its implications for paleostudies) is discussed in depth in section 4.3 (3rd paragraph). At the deepest station (with the highest BWO content) Mn-oxides are accumulating in the sediments. As BWO content is relatively high, Mn is trapped and not able to escape into the water column, which may possibly occur at station 6 and 8. In addition it is possible that Mn-oxides are transported along slope, hence accumulating at the deeper and more ventilated areas. Although at each station some Mn-reduction was taking place, as shown by subsurface peaks in Mn, the pool of total dissolved Mn is likely to be related to availability of Mn-oxides in the sediment.*

page 10:
l. 11" deeper in the sediment where higher Mn conc. are present": I do see the increase in Mn with depth only at station 10, not the others, so this statement in my opinion cannot be drawn. "a clear increase in foraminiferal Mn is observed as well": In this case, it would be very valuable to show a regression of foraminiferal Mn to porewater Mn to underline this statement.

*RESPONSE: Indeed the largest increase in the pore water Mn with sediment depth is observed at station 10. At the other stations Mn-reduction is more limited, however, a small dissolved  Mn- peak is also present at station 6 (at depth of 0.8 cm) and 8 (at depth of 1.3 cm), implying that Mn- reduction occurs in the sediment. Nevertheless, at station 10, where a clear peak in Mn is present this is also reflected in the foraminiferal Mn/Ca ratios, as we see very low concentrations in the surface dwelling E. batialis and high concentrations in deep living N. labradorica.  The sentence is now modified to make clear that this refers specifically to station 10. In addition, "clear" is omitted. Unfortunately species-specific regressions of foraminiferal Mn to pore water Mn are not possible due to limited pore water data.*

page 11:
section (4.3). should be revised- at the moment, the paleographic implications from the measurements presented here (Mn/Ca in foraminifera and Mn in porewaters), should be the main focus in addition to comparison to literature values (this part is included). However, the present version discusses the relevance of the Troxchem model at length in addition the Mn redox chemistry, however, only very little focus lies on the paleo implications of this study. Please move the discussion of the TROXCHEM model and the redox chemistry into a different paragraph.

*RESPONSE: Section 4.3 has been revised and the main paragraph dealing with the TROXCHEM model has been moved to section 4.2. However, we have decided to keep the discussion on Mn-redox chemistry as part of 4.3 as it has direct implications for the application of Mn/Ca down core. Our data shows that foraminiferal Mn/Ca is not only reflecting redox conditions but is also influenced by availability of Mn-oxides and hence Mn to be potentially released upon reduction. The availability of Mn-oxides, and hence the MnOx-reduction potential of the sediment, as shown by our results, is recorded in the foraminifera along our study transects. Therefore, paleoceanographic studies should take into account changes in the supply of Mn-oxides as well as changes in sediment oxygenation, as the former is also an important parameter in regulating pore water Mn-concentrations.*

Furthermore, I am having a hard time to discern the key messages of the study wrt to what influences foraminiferal Mn/Ca. I agree with conclusion, that deeper fauna displays higher Mn/Ca, and that the deeper species must be calcifying under dysoxic conditions, but from the data presented I am having a hard time to see that "Mn incorporation" reflects
*RESPONSE: conclusions have been modified*
1)bottom water oxygenation (where is the data- regressions/ statistics and or figures? e.g. regression of foram Mn and BWO) representing this?
*RESPONSE: A statistically significant correlation was observed between BWO and Mn/Ca ratios in B. spissa (Results: section 3.4). Also statistically significant increases in the Mn/Ca ratios were observed along the study transect for Uvigerina spp. and B. spissa (Results: section 3.4).  No regression analyses were carried as there is not sufficient pore water data (i.e. each species was not present at all three sites with available pore water data) to support this.*

2) Mn redox chemistry (where is the data? regressions? statistics)
*RESPONSE: Mn pore water data was collected at three stations only and maximum of 2 taxa were measured at stations with pore water Mn-data. Hence unfortunately there is not sufficient Mn pore water data to carry out such analyses. Conclusions have now been modified to reflect this.*

3) no ontogenetic influence (as argued above, it could be that interspecies variability masks this, since on most specimen, only 2,3 chambers are measured. However, I am positive that data can be easily presented in a revised version to be able to make this statement.
*RESPONSE: the reviewer likely refers to intraspecies variability, not interspecies variability. It is true that most specimens of Uvigerina spp. were measured two or three times, however, this is not true for B. spissa, which was measured 4 times in 14 out of 23 specimens. In addition, E. batialis was measured twice 5 times and once 6 times.  None of these specimens showed systematic, statistically significant ontogenetic trends (now added to the appendix 1). Therefore, we can further conclude that no ontogentic trends were observed in our data. This is now clarified in results section 3.2. with a reference to Appendix 1.*

*minor comments:*
page 1
l.16: calcium carbonate tests
*RESPONSE: ok, done*
l.19: define BWO or spell out; what are differences exactly?
*RESPONSE: ok, done*
l.20: where is this entangling happening in manuscript?
*RESPONSE: changed "further resolving"*

l.24: At each station, Mn/Ca (omit "the")
*RESPONSE: ok, done*
also Mn/Ca is a ratio of concentrations, not a concentration
*RESPONSE: ok, changed to ratio*
l.31: the forams are not the tools, but carry the proxy -> rephrase
*RESPONSE: changed to proxies used in paleoceanographic studies.*
l.32: has a high…
*RESPONSE: ok, done*

page 2
l.4: have been shown to reflect carbonate chemistry (omit "the")
*RESPONSE: ok, done*
l.18: are oxygenated and sediments are anoxic… add "and sediments are anoxic"
*RESPONSE: changed to "In sediments, where bottom waters and surficial sediments are oxygenated and deeper sediments are anoxic…"*
l.27: omit "the" before shallow
*RESPONSE: ok, done*
l.28: than not then
*RESPONSE: ok, done*
l.32: why 33 (random?) also omit "the" before foraminifera and change to foraminiferal
*RESPONSE: 33 is not random this is the lowest BWO content along the study transect, so BWO was always higher than this. "The" is now omitted.*

page 3
l.17: change to sth. like this as it is confusing otherwise: "At each site, three separate…"
*RESPONSE: "Separate cores were collected for pore water- and foraminiferal analyses, and oxygen profiling, all of which were derived from the same multicore cast."*
l.19: company that produced CTD (seabird?), what is the error of the oxygen microsensor? Is it also called a "micro"sensor when it is attached to a CTD?
*RESPONSE: The CTD is SBE9plus (Sea-Bird Electronics, S/N 860) and it was equipped with SBE3 thermometer (S/N 4378), SBE4 conductivity sensor (S/N 3307) and SBE43 oxygen sensor (S/N 0781). The details of the equipment are now added into the manuscript (section 2.2). "Oxygen microsensor" replaced with "oxygen sensor". The accuracy specifications of the oxygen sensor are typically within 2% of true value*

l.25: Whole sample centrifuged or subsample?
*RESPONSE: Changed to "Sediment samples were centrifuged…" In general the whole samples were centrifuged. In case of the deepest sediment intervals where the slice thickness was 2 cm, some sediment may have been disregarded.*
l.30: how much HCl was added? final conc.? What samples were used for storage? Were they acid cleaned?
*RESPONSE: Text modified "Samples for pore water elemental analyses were acidified with suprapur HCl 37% (10µl per ml of sample) and subsequently stored at 4°C until analyses at Utrecht University."*
*The foraminiferal samples were not cleaned other than was indicated in the original manuscript.*

page 4
l.1: I assume cps were measured and then converted to conc. via a calibration curve for those elements measured on the ICPMS? What wavelengths were measured in the OES? Which elements were measured on which machine? Which isotopes were measured on the ICPMS?
*RESPONSE: Sentence modified to "Seawater elemental concentrations of $^{55}$Mn were measured with an inductively coupled plasma-mass spectrometer (ICP-MS, ThermoFisher*

*Scientific Element2-XR)." Part about the OES is deleted as only ICP-MS data is reported in this article.*

l.5 - 12: As I am unfamiliar with the methods and the custom built incubation chamber please provide a few more details to clarify:
I assume the subsample taken with the syringe was analyzed? Stabilization of what? temp. and oxygen? How were the fluctuations in oxygen conc. assessed? Were the stabilization times similar between cores (ca. 9hrs)? Were the oxygen profiles taken continuously or at certain depths?
*RESPONSE: The O2 profiles have been published previously in Fontanier et al. (2014) with details of the employed methods. A citation has been added here.*
*Each time a core was left to stabilize under insitu $O_2$ and temperature conditions for 9hrs (as these parameters are likely to change during core recovery). $O_2$ conditions were monitored with a microsensor, with no syringe being used. $O_2$ profiles were made at 100μm resolution.*

l.13: Change title so it is more precise: e.g. "Foraminifera: sampling an elemental concentrations"
*RESPONSE: changed to "Foraminifera: sampling and elemental composition"*

l.14 et al.
*RESPONSE: ok done*

l.17: Plummer slides? Are they micropaleoslides?
*RESPONSE: ok done*

l.30: So if the crater is 80μm I assume all foraminiferal chambers measured are bigger than that to make sure, that only one chamber is ablated per measurement?
*RESPONSE: The word "generally" was added. In general always single chamber was ablated, however, it can not be completely excluded that in few rare cases two chambers were ablated at once. This would mainly concern small individuals of B. spissa, which has very small older chambers.*

page 5
l. 26: NFHS: has the homogeneity of this standard been published somewhere? Were JCP21 MACS3 and NFHS all used as the form of pressed powder tablets?
*RESPONSE: Yes: all these $CaCO_3$ powders were pressed into tablets. The relative standard deviation in element/Ca based on multiple measurements on the NFHS is comparable to that of other standards (Mezger et al., 2016). A reference to this study is added (section 2.5 5th paragraph)*

l.30: I assume seawater= porewater? where is DMn reported?
Knowing the good quality of data that usually is published from the Utrecht setup used, I assume that the methods have been written up by two different co-authors, I would strongly encourage the authors to rewrite section 2.5 so that the same details are given for both setups used.
*RESPONSE: DMn values are all given now in Appendix 2.*
*Yes, seawater changed to pore water.*
*We have carefully checked section 2.5 to check for consistency and modifications have been made. However, the differences in setups and controls of the ICP-MS's used inherently cause the descriptions to differ. For example, tuning of the quadrupole and SF-ICP-MS differ and also the way they cycle through the elements analysed differs fundamentally. Still, all relevant parameters are described in each of the paragraphs.*

page 6
l.6: what exactly is pore water chemistry? which parameters?
*RESPONSE: Sentence modified to "Pore water chemistry, including dissolved oxygen, nitrate, ammonium and manganese, was measured at sites 6, 8 and 10 (Figure 2)."*

l.8: in-sediment depth? what depth is this?
*RESPONSE: Sentence modified to" In all cores, nitrate was rapidly depleted within surficial sediments"*

l. 23: App. 1 is missing, I contacted the first author for App. 1, the excel file I received looks like there was mostly 2-3 chambers measured on each specimen, so that I doubt that this is enough to support that "there is no correlation between shell size and Mn/Ca" as it could be that interspecies variability masked potential ontogenetic trends in Mn/Ca, if only 2 or 3 chambers were measured on one specimen. I would encourage the authors to provide a figure in the appendix to demonstrate intra-species variability and also to calculate inter- versus intra-species variability for all species studied and provide data in a table. Also I do not see statistical analyses in App. 1 (L. 23: "The statistical analyses were carried out on all data (App.1)").
*RESPONSE Appendix is now added, containing all raw data. This is labeled now as appendix 2. In addition, appendix 1 is now supplied where profiles of B. spissa and E. batialis are shown, based on Mn/Ca ratios in single specimens.*
*Pearson correlation coefficients were calculated, however, all correlations were insignificant with two-tailed significance always being >0.05. This is added to results section 3.2.. Please also consider the response on this same topic earlier in the review.*

page 7
l. 28: lowest average (?) Mn/Ca values
*RESPONSE: Nina Keul was contacted regarding this comment, as it was not clear to the authors what she originally meant with it. In her response she stated " I was wondering whether average shell Mn/Ca is in station 6 the lowest for that species or whether it is actually lower in shells from the same species in station 8, where porewater Mn was higher? (Sorry for the kryptic comment...) If that's the case it should be discussed somewhere.*
*Our response to this comment: The average Mn/Ca ratio at station 6 is 30,1 and at station 8 30,6, so it is little bit higher at station 8 than 6, as should be due to slightly higher pore water Mn-content at station 8. However, the difference is not statistically significant. Based on this comment, it was noted that the standard error is missing in Figure 8 for Uvigerina spp (station 8) this is now added to the figure. For station 7 the error bar is so small (1,7) that it is hidden under the data label, and thus hardly visible.*

page 8
l.16: "excluded from data": show also in exemplary profile (see comment above)
"Due to the nature of the specimens..." does this refer to the fact that living foraminifera most likely do not have diagenetic coatings or some other factor?
*RESPONSE: representative laser ablation profiles are now provided (figure 2). Indeed as specimens were very recent, diagenetic coatings are unlikely. This is now added into the sentence.*

page 9:
please add references to figures and tables (also the "new" one with the Mn/Ca and DMn values)
*RESPONSE: new figure with $D_{Mn}$ made and referred to. Values also given in Appendix 2.*

l.15/16: bimodal distribution - which species here shows a bimodal distribution?

*RESPONSE: B. spissa at station 7 and Uvigerina spp. station 7 and 8 (Fontanier et al. 2014). This is now added into the sentence.*

l. 26: delete "are"
*RESPONSE: Modified to "…it seems that deep infaunal foraminifera, based on their Mn incorporation, are calcifying…"*

*page 10:*
l. 20: fluxes must still be relatively
*RESPONSE: ok, done*

l.32 remove "study" at end of sentence
*RESPONSE: ok, done. This section has also now moved up into the start of section 4.3*

page 11:
l.6: fig 2 not fig1
*RESPONSE: reference to figure corrected.*

Last paragraph: good discussion of Mn redox chemistry and availability, but maybe move upin the manuscript, as it is in general relevant for the incorporation of Mn into foraminifera and not necessarily part of the "paleo implications only".
*RESPONSE: As outlined earlier in the response to the review, we feel that this has direct implications for paleo studies as it shows that the Mn/Ca ratios in foraminfera do not only depend on oxygenation, or redox chemistry, but also on supply of manganese oxides in sediment. Therefore, we have kept this section as part of 4.3.*

---END OF REVIEW 1-----

**REVIEW 2: Anonymous referee#2**

The manuscript "Benthic foraminiferal Mn/Ca ratios reflect microhabitat preferences" by Koho et al. presents new data on the link between pore water Mn concentrations which are related to dissolved oxygen content, and benthic foraminiferal Mn/Ca. Mn/Ca is receiving a lot of attention recently as it may be a suitable proxy to reconstruct past dissolved oxygen concentrations in the water column/pore water. Using several different species and linking the data with pore water measurements has resulted in a very nice dataset, which partly provides evidence for existing ideas but also points out some issues that still exist. Especially the discussion on these possible issues could still use some more extensive consideration as described below in detail. But in general, the manuscript is well-written, easy and clear to follow, and definitely fitting within the scope of Biogeosciences. I recommend that this manuscript is suitable for publication after moderate revisions have been made.
*RESPONSE: The authors thank reviewer#2 for the time spent reviewing the manuscript and the positive feedback on the study.*

My main issue is that I feel that the discussion on the part where pore water Mn/oxygen and Mn/Ca in the forams are not fitting, can be explored further. Currently, it is partly contradicting, i.e. living labradorica and fimbriata were found at 0-1 cm but are generally deeper-living species (unless maybe in conditions where the bottom water is already close to anoxic), so that would imply habitat migration. But then the lack of a trend in Mn/Ca in the chambers would indeed point to no migration. In station 8, both species have the highest Mn/Ca again and are deepest, but there is no Mn in the pore water. So under the anoxic conditions all the available Mn has either diffused upwards

when reduction took place or it precipitated as MnCO3. How then can the forams have high Mn/Ca? For me this either means that they did migrate and picked up the Mn at a shallower depth; or that pore water oxygen and thus Mn are changing through the seasons, having higher pore water Mn when the forams calcified (assuming they were not calcifying at the moment of collection); or finally that the test Mn/Ca is biased by MnCO3 precipitation. You did write that contamination on in- and outside bits (high Al and or Mn) was discarded, but it would be interested to know if especially in these deep station 8 forams there was indeed a Mn-coating. Because if a coating forms, crystals may as easily form somewhere inside the test to bias the bulk Mn/Ca.

*RESPONSE: A discussion dealing with the influence of foraminiferal migration has been added to section 4.2, 4th paragraph. Although, no systematic ontogenetic migration was observed, it is possible that foraminifera move in the sediment during their life. As the study of Fontanier et al (2014) shows some deep infaunal foraminifera, including N. labradorica, are also found in low abundances at surface sediments, although their ALD and maximum density is deeper. The migration of intermediate and deep infauna foraminifera in sediment may also explain the larger scatter seen in the Mn/Ca ratios of these species, whereas specimens from the surface dwelling E. batialis had relatively similar Mn/Ca ratios.*
*Two typical laser ablation profiles are now given in new Figure 2. In all cases any contamination was excluded, and specimens with relatively high final Al content were discarded prior to further statistical analyses. In addition, all specimens used in the study were rose Bengal stained and therefore alive at the time of sampling, or dead very recently. Hence, high Mn/Ca ratios due to diagenetic coatis can be neglected.*

Even though that in general the relation between oxygen and Mn/Ca seems to follow the expected trends, the species-specific correlations are not very good or non-existing. What do you think could be the reason for that? How could the impact of habitat migration be determined? Seasonality may be resolved of course by extra sampling, which is always welcome. As a side note, I do like to point out that it would have been great to have had pore water profiles for stations 7 and 9 too.
*RESPONSE: We suspect that part of this is due to lack of pore water data. Unfortunately the team responsible for the pore water analyses could handle only three stations within the time available. The oxygen free sampling (slicing), porewater extraction and subsampling of porewaters is very labor intensive, resulting in the sampling of pore waters from 3 stations only. In addition, foraminiferal migration and seasonal changes in pore water could explain some of the discrepancies seen in the data. This is now discussed in section 4.2 4th paragraph.*

Minor Comments: 2.1 add some of the main currents and water masses to figure 1.
RESPONSE: The pathway of Tsugaru warm current (based on Oguma et al., 2002) is now added to the Figure 1. As the position of the currents shifts during the year, we do not feel confident placing the Oyashio current on the map. The dysoxic water mass is indicated in figure 1. As the North Pacific intermediate Waters (NIPW) mixes gradually with saline Deep Pacific Water (DPW), entering this area between a water depth of 800-3000 m, it is not possible to accurately place the water mass boundaries on the Figure 1C.

p.4, 16: part of the previously mentioned loop of possible explanations why not everything fits. Could it be that some of the deeper specimens in the anoxic sediment are stained despite being dead? They would still classify as recently-alive, but that may be enough to have them buried a couple of cms.
*RESPONSE: It is true that foraminifera could be recently dead, and hence buried. However, it should be noted that the pore water profiles provide a snap shot in time, and hence the*

*conditions may have been slightly different at the time when the foraminifera calcified. This is now discussed in section 4.2 4th paragraph.*

p.5, 6: Mg? Mg/Ca data would of course also be interesting to present. But to stick to redox elements, were any other redox elements like Fe or U analyzed?
*RESPONSE: Focus of this study is Mn/Ca. Fe was analyzed but the data does not seem good enough for any robust inferences. Measuring Fe with laser ablation is challenging due to interferences with other masses (e.g Ar-O). This can be overcome by measuring the samples at intermediate resolution with the SF-ICP-MS, but is not possible with the quadrupole.*
*Uranium was measured. However, previous studies have also shown it to vary with carbonate saturation (e.g. Raitzsch et al. 2011 G³). Thus, adding U-data to the manuscript would reduce the focus of the paper.*

p.5, 26: Internal reproducibility is good, but how was the comparison between both lasers?
*RESPONSE: As shown earlier, these systems provide consistent results (De Nooijer et al., 2014a). We have added this reference to the manuscript (Section 2.5. paragraph 3), which is also reflected by the similarity in precision/ accuracy of the ablated standards at both instruments.*

p.6, 23: shell size; can a trend in different chambers automatically be related to shell size? I am not sure if this is a correct way of naming it.
*RESPONSE: Chamber number refers to the ontogenetic stage, F- being most recent, F-1 being penultimate chamber and so on.*

p.7, 29: rations, delete n
*RESPONSE: mistake not found on p. 7 line 29 or elsewhere in the document.*

p.10, 20: where the TROXCHEM, add the; still be, add be
*RESPONSE: done*

p.12, 2: foraminferal
*RESPONSE: done*

table 2: change comma's for decimals to points.
*RESPONSE: done*

Figure 1: add currents and watermasses
*RESPONSE: The pathway of Tsugaru warm current (based on Oguma et al., 2002) is now added to the Figure 1. As the position of the currents shifts during the year, we do not feel confident placing the Oyashio current on the map. The dysoxic water mass is indicated in figure 1. As the North Pacific intermediate Waters (NIPW) mixes gradually with saline Deep Pacific Water (DPW), entering this area between a water depth of 800-3000 m, it is not possible to accurately place the water mass boundaries on the Figure 1C.*

Figure 5 caption: in indicated, change to is
*RESPONSE: done*

Figure 6 caption: this is exactly the same as the one for figure 5, which I assume should not be the case.
*RESPONSE: done*

---END OF REVIEW 2-----

[revised manuscript text omitted]

Karoliina Koho 6/4/2017 14:27

Karoliina Koho 6/4/2017 14:27

Karoliina Koho 6/4/2017 14:27

Karoliina Koho 6/4/2017 14:27

Karoliina Koho 6/4/2017 14:27

Karoliina Koho 6/4/2017 14:27

Karoliina Koho 6/4/2017 14:27

Karoliina Koho 6/4/2017 14:27

Karoliina Koho 6/4/2017 14:27

Karoliina Koho 6/4/2017 14:27

Karoliina Koho 6/4/2017 14:27

Karoliina Koho 6/4/2017 14:27

Karoliina Koho 6/4/2017 14:27

Karoliina Koho 6/4/2017 14:27

Karoliina Koho 6/4/2017 14:27

Karoliina Koho 6/4/2017 14:27

Karoliina Koho 6/4/2017 14:27

Karoliina Koho 6/4/2017 14:27

[Figure]

[Figure]

Karoliina Koho 6/4/2017 10:24

[revised manuscript text omitted]

---

## Author Response (AR2)

Helsinki 23 May 2017

Dear Editor,

Hereby we submit all files required for the production process of the final version of our manuscript titled "Benthic foraminiferal Mn/Ca ratios reflect microhabitat preferences". We have now addressed all the remaining points raised by the reviewer (Nina Keul). Below you can find the point-by-point response to each of her comments.

Thank you for considering our manuscript for publication in the journal *Biogeosciences.*

Best wishes,
Karoliina Koho (on behalf of all co-authors)

Review by Nina Keul

The authors have incorporated the suggestions raised by me and another reviewer into the manuscript, so that it is now more concise and transports the main conclusions in a logical manner that is supported by data. Before the manuscript is ready for publication, substantial editing of the paper has to be carried out, as there are still many small mistakes and inconsistencies, e.g. space between the numerical value and unit symbol, abbreviations are inconsistent (e.g. m or meter), spelling out of numericals is inconsistent, etc.
For instance page 3: l. 8: 200m, l. 10: 700 - 1500 meters, l. 6: three, l. 18: 3
*Author response: We thank Nina Keul for the second thorough review of the manuscript. The manuscript has now been edited as suggested by the reviewer.*
*-All whole numbers between zero and ten have been spelled out. All numbers above ten, or with decimals, have been indicated with numericals.*
*-A space has been added always between numerical value and a symbol*
*-Abbreviations are now consistent. If number spelled then also symbol is spelled out. If number in numerical a abbreviation (or a symbol) is used.*

Other minor revisions:
-Figures:
Panel b Fig. 1, indicate in caption what black and white triangles indicate, Panel c of Figure 1 did not reproduce the grey shading on two color printers I tried, maybe that can be changed, maybe also indicate the depths of the dysoxic waters in the caption.
*Author response: Caption modified. Information on triangles added as well as the depth of the dysoxic water mass. The shading used to indicate the dysoxic water mass has also been made darker.*

Fig 4.: Indicate in boxplots the percentages (e.g. whiskers are 1,5×IQR I assume?) what did you define as outliers (3x IQR?).
*Author response: caption modified and details added.*

-Appendix: The appendix table 2 needs some formatting e.g. column 5, the u in umol should be replace by the Greek letter, commas should be replace by points.
*Author response: ok added, commas replaces by points.*
Please also include a table with average values of Mn/Ca and DMn (referring to Figs. 6 and 7), not only the raw data.
*Author response: the average values are already given in figures 6 and 7 and therefore it seems unnecessary to add yet another table into the main manuscript. Howevere, a table has been added to the appendix 2 where readers who are interested in seeing the exact numbers can easily find them.*

I congratulate the authors on such an interesting manuscript and remain available for further questions.

Best,
Nina Keul